# SeTAR: Out-of-Distribution Detection with Selective Low-Rank Approximation

**Yixia Li**[1][*], **Boya Xiong**[2][*], **Guanhua Chen**[1][†], **Yun Chen**[3][†]

[1]Southern University of Science and Technology
[2]Shanghai University of Finance and Economics
[3]MoE Key Laboratory of Interdisciplinary Research of Computation and Economics,
Shanghai University of Finance and Economics
`liyixia@me.com, xiongboya@163.sufe.edu.cn`
`chengh3@sustech.edu.cn, yunchen@sufe.edu.cn`

## Abstract

Out-of-distribution (OOD) detection is crucial for the safe deployment of neural networks. Existing CLIP-based approaches perform OOD detection by devising novel scoring functions or sophisticated fine-tuning methods. In this work, we propose SeTAR, a novel, training-free OOD detection method that leverages selective low-rank approximation of weight matrices in vision-language and vision-only models. SeTAR enhances OOD detection via post-hoc modification of the model's weight matrices using a simple greedy search algorithm. Based on SeTAR, we further propose SeTAR+FT, a fine-tuning extension optimizing model performance for OOD detection tasks. Extensive evaluations on ImageNet1K and Pascal-VOC benchmarks show SeTAR's superior performance, reducing the relatively false positive rate by up to 18.95% and 36.80% compared to zero-shot and fine-tuning baselines. Ablation studies further validate SeTAR's effectiveness, robustness, and generalizability across different model backbones. Our work offers a scalable, efficient solution for OOD detection, setting a new state-of-the-art in this area.

## 1 Introduction

The task of out-of-distribution (OOD) detection (Hendrycks & Gimpel, 2017; Ming et al., 2022) aims to identify whether input data comes from an unknown distribution. It has garnered significant attention in the machine learning community recently (Hendrycks et al., 2020; Xu et al., 2021; Miyai et al., 2023a). While machine learning models are trained with supervised in-distribution (ID) data, they often struggle to generalize to OOD data encountered in real-world applications (Emmott et al., 2016) like autonomous vehicles and healthcare. These OOD samples pose challenges as they are not represented in the training data. Consequently, OOD detection plays a crucial role in developing reliable and trustworthy machine-learning models suitable for real-world deployment (Bai et al., 2023). It allows models to filter out and reject these awkward inputs effectively, and enables the use of curated and labeled OOD samples to further train for a more robust model in the wild.

Previous research has primarily focused on detecting OOD instances in either visual (DeVries & Taylor, 2018; Liang et al., 2018; Hendrycks et al., 2022) or textual data (Hu & Khan, 2021; Zheng et al., 2020; Zhou et al., 2021). Recently, significant progress has been made in multimodal tasks like multimodal retrieval (Li et al., 2023; Caesar et al., 2018) and image classification (Yu et al., 2022), thanks to vision-and-language pretrained (VLP) models like CLIP (Radford et al., 2021). More recent

---

[*]Equal Contribution.

[†]Corresponding Authors.

38th Conference on Neural Information Processing Systems (NeurIPS 2024).

studies have explored OOD detection with CLIP, grouped into zero-shot methods (Fort et al., 2021; Ming et al., 2022; Miyai et al., 2023b) and finetuning-based methods (Ming & Li, 2023; Tao et al., 2023; Miyai et al., 2023a). However, the zero-shot methods suffer from suboptimal performance due to potential domain gaps with ID downstream data. On the other hand, finetuning-based methods carry the risk of deconstructing the intricate representations learned by CLIP which requires a meticulously designed training strategy. Sparsification-based approaches (Sun et al., 2021; Djurisic et al., 2023) have demonstrated potential in OOD detection within CNNs, leveraging the assumption that ID and OOD samples produce distinct activation patterns. Nevertheless, their effectiveness diminishes in large-scale pre-trained models such as CLIP, where activation differences become more subtle, thereby limiting their applicability primarily to models fine-tuned on downstream ID-domain datasets.

In this work, we propose SeTAR, a training-free and effective OOD detection method by selective low-rank approximations. Low-rank approximation is to approximate a given matrix by finding a lower-rank matrix that closely resembles the original matrix. Previous research has demonstrated that using low-rank approximation matrices can achieve comparable performance to full parameters in various scenarios, as observed in tasks such as large language model (LLM) fine-tuning (Hu et al., 2022) and model pruning (Hajimolahoseini et al., 2021). These approaches typically preserve the same rank across different low-rank approximation matrices. In our work, we demonstrate that it is possible to significantly enhance the performance of OOD detection by selectively manipulating the weight matrices in the CLIP model, including the choice of weight matrices and the ratio of singular vectors to be reduced. Specifically, we propose a simple top-to-bottom and image-to-text greedy search algorithm to manipulate $W_{up}$ in the CLIP model. Our method applies to various model backbones and does not require any additional training or new parameters. Building upon SeTAR , we further demonstrate its effectiveness for fine-tuning initialization, referred to as SeTAR+FT.

We conduct extensive evaluations and achieve state-of-the-art performance on common OOD detection benchmarks for CLIP, including the ImageNet1K and Pascal-VOC benchmarks. Compared to vanilla MCM and GL-MCM, SeTAR with the CLIP backbone reduces relatively FPR95 by 9.5% and 12.0% on average across two benchmarks, respectively. When further integrate fine-tuning into SeTAR, SeTAR+FT outperforms the state-of-the-art fine-tuning baselines LoCoOp (Miyai et al., 2023a) and LoRA (Hu et al., 2022). Moreover, we perform a comprehensive ablation study and analysis to verify and understand SeTAR. In summary, our key results and contributions:

1. We propose SeTAR, a simple yet effective OOD detection method based on selective low-rank approximation. It is training-free as it only performs post-hoc modification to weight matrices. SeTAR applies to a variety of scoring functions and model backbones. It can be readily integrated with existing zero-shot OOD detection methods.

2. We further extend SeTAR to SeTAR+FT, which demonstrates the effectiveness of SeTAR in improving the performance of finetuning-based OOD detection methods and achieving new state-of-the-art results.

3. We extensively evaluate SeTAR and SeTAR+FT across a diverse set of OOD detection tasks. It consistently outperforms baseline methods and establishes new state-of-the-art results on CLIP-based OOD detection benchmarks. On ImageNet1K, SeTAR achieves an AUROC of 91.32% with CLIP backbone and GL-MCM score. The score further increases to 92.31% when combined with the finetuning-based detection method.

4. We perform comprehensive ablation studies and empirical analyses to verify and understand SeTAR. We hope that this work will shed light on future explorations on in-depth understanding of the SeTAR method.[3]

## 2 Preliminaries

**CLIP Architecture**  The CLIP model (Radford et al., 2021) comprises an image encoder $E^v(\cdot)$ and a text encoder $E^t(\cdot)$, aligned via contrastive learning on web-scale image-text pairs. We focus on CLIP-ViT, where the image encoder is a Vision Transformer (ViT). Each ViT layer includes a multihead self-attention sublayer and a feed-forward sublayer. In the self-attention module, the hidden state is projected into different spaces using learnable parameters $W_q, W_k, W_v$. The outputs are

---

[3] Code are available at `https://github.com/X1AOX1A/SeTAR`.

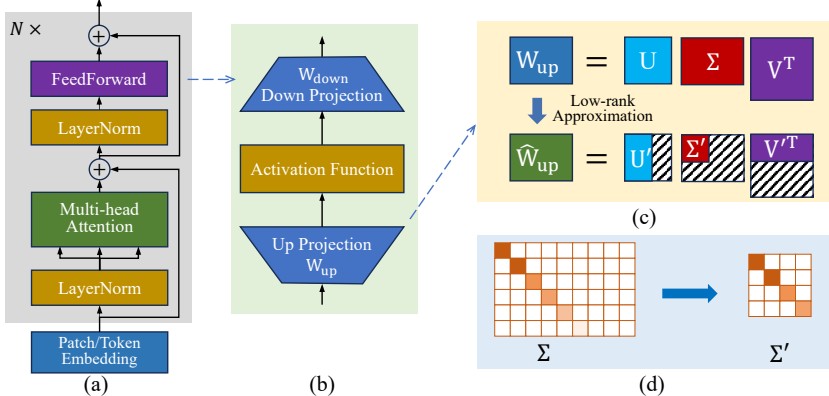

Figure 1: The overview of SeTAR. (a) The structure of the CLIP image and text encoder. (b) The details of the feed-forward sublayer. (c) For each encoder layer, we replace the $W_{up}$ weight matrix with its low-rank approximation $\widehat{W}_{up}$. (d) The illustration of $\Sigma$ before and after low-rank approximation. More details are in Section 3.1.

concatenated and projected back with another linear matrix $W_o$. The feed-forward module projects the hidden state into a wider space using $W_{up}$ and then back with $W_{down}$ after a non-linear activation (Figure 1). Given the similarity between the image and text encoder layers, we adopt consistent notations for the linear matrices in both. Each encoder also includes a linear projector $W_p$ to map their representations into a shared space for contrastive learning.

**Zero-shot OOD Detection with CLIP**  Zero-shot OOD detection aims to separate ID and OOD data without an ID training dataset. Given the CLIP, the ID classes are defined by the classification task of interest, which differs from the classes used in CLIP pretraining. Accordingly, OOD is defined as classes not belonging to any ID class, making the OOD detector a binary classifier. MCM (Ming et al., 2022) and GL-MLM (Miyai et al., 2023b) are two zero-shot CLIP-based OOD detection methods. Formally, let $\mathbf{x}$ be the test image and $\mathcal{T}_{in} = \{\mathbf{y}_c\}_{c=1}^{K}$ be the set of text prompts containing $M$ ID class labels (e.g., "a photo of a [CLASS]"). The image is segmented into $l$ image patches $\mathbf{x} = (x_1, ..., x_l)$. Following CLIP, we add [cls] before the image patches and use the output of [cls] from the visual projector $W_p$ as the global image feature ($h^v \in \mathbb{R}^d$). The outputs of other patches are projected by the visual projector as the local image features ($\mathbf{p}^v = (p_1^v, ..., p_l^v) \in \mathbb{R}^{l \times d}$). For the text prompt $\mathbf{y}_c \in \mathcal{T}_{in}$, we add an additional [eos] after the text tokens and use the output feature of [eos] from the textual projector $W_p$ as the concept feature of ID class $c$ ($h_c^t \in \mathbb{R}^d$).

The label-wise image-concept matching (IWIC) score measures how well a test image $\mathbf{x}$ aligns with a concept $\mathbf{y}_c$, using either global or local features. The global IWIC score $s_c^G(\cdot)$ is the cosine similarity between the global image feature $h^v$ and the concept feature $h_c^t$: $s_c^G(\mathbf{x}) = \cos\_sim(h^v, h_c^t)$. The local IWIC score $s_c^L(\cdot)$ is the max-pooled cosine similarity between image patch features $p_i^v$ and the concept feature $h_c^t$: $s_c^L(\mathbf{x}) = \max_i \cos\_sim(p_i^v, h_c^t)$. The MCM and GL-MCM scores are defined as:

$$S_{\text{MCM}}(\mathbf{x}) = \max_c \frac{e^{s_c^G(\mathbf{x})/\tau}}{\sum_{c=1}^{K} e^{s_c^G(\mathbf{x})/\tau}}, \tag{1}$$

$$S_{\text{GL-MCM}}(\mathbf{x}) = S_{\text{MCM}}(\mathbf{x}) + \max_c \frac{e^{s_c^L(\mathbf{x})/\tau'}}{\sum_{c=1}^{K} e^{s_c^L(\mathbf{x})/\tau'}}, \tag{2}$$

where $\tau$ and $\tau'$ are the temperature hyperparameters. MCM only uses global image features, while GL-MCM additionally considers local image features. For ID data, both MCM and GL-MCM scores will be matched to one of the concept features with a high score; and vice versa. As a result, our OOD detection function can be formulated as:

$$G(\mathbf{x}) = \begin{cases} 1 & S(\mathbf{x}) \geq \lambda \\ 0 & S(\mathbf{x}) < \lambda \end{cases}, \tag{3}$$

where $S(\mathbf{x})$ is either the MCM or GL-MCM score, $\lambda$ is the threshold value. By convention, $G(\mathbf{x}) = 1$ represents the ID class and $G(\mathbf{x}) = 0$ indicates the OOD class. The $\lambda$ is chosen so that a high fraction of ID data (e.g., 95%) is above the threshold. We follow previous work (Miyai et al., 2023a) to use either MCM or GL-MCM score for OOD detection in this work.

# 3 Method

We introduce SeTAR, a training-free and effective technique for improving OOD detection performance (see Figure 1). Our key idea is to perform post-hoc modification to CLIP weight matrices by selectively replacing them with their low-rank approximations. It is complementary to existing CLIP-based zero-shot OOD detection methods and could be further extended to finetuning-based methods, which we term as SeTAR+FT.

## 3.1 OOD Detection with Selective Low-Rank Approximation

**Low-Rank Approximation**   Given a linear matrix $W \in \mathbb{R}^{m \times n}$, its Singular Value Decomposition (SVD) is denoted as $W = U\Sigma V^{\top}$, where $U = [u_1, u_2, \cdots, u_m] \in \mathbb{R}^{m \times m}$, $V = [v_1, v_2, \cdots, v_n] \in \mathbb{R}^{n \times n}$, and $\Sigma \in \mathbb{R}^{m \times n}$ is a matrix whose entries are all zero except for the singular values of $W$. These singular values appear in decreasing order on the diagonal (i.e. $\sigma_i^{\downarrow}(W)$). The SVD of $W$ can be reformulated as in Equation 4. Given a hyperparameter $r \in \mathbb{N}^{+}$, a rank-$r$ approximation of $W$ is matrix $\widehat{W}$ that minimizes $\|W - \widehat{W}\|_2$ and satisfies $\text{rank}(\widehat{W}) \leq r$. The optimal solution of this problem $\widehat{W}$ is provided by Eckart–Young–Mirsky theorem (Low-Rank Approximation, 2024) using Singular Value Decomposition (see Equation 5).

$$W = \sum_{i=1}^{\min(m,n)} \sigma_i^{\downarrow}(W)u_i v_i^{\top}, \tag{4}$$

$$\widehat{W} = \sum_{i=1}^{r} \sigma_i^{\downarrow}(W)u_i v_i^{\top}. \tag{5}$$

In this work, we will use the term **minor singular components** to refer to entries in the SVD corresponding to small singular values. These components are removed in low-rank approximation. The term of **principle singular components** is used to refer to entries in the SVD corresponding to large singular values. These components are kept in a low-rank approximation of the matrix.

**OOD Detection with Selective Low-Rank Approximation**   SVD-based weight pruning, particularly in noise-prone layers, can substantially reduce a network's sensitivity to minor perturbations, leading to enhanced stability and robustness (Yao et al., 2024). This stability is crucial for OOD detection, as it ensures the model's reliable performance across a wide range of inputs. Building on this, we propose a method to improve OOD detection by selectively applying low-rank approximation to weight matrices. By decomposing a weight matrix $W$ into its singular values and vectors, we can identify and retain the principle singular components that significantly contribute to the model's performance. This approach ensures that the essential features of $W$ are preserved while discarding the less critical minor singular components. Given a weight matrix $W$ in CLIP (e.g., $W_{\text{up}}$ or $W_{\text{k}}$), we replace the matrix with its low-rank approximation part $\widehat{W}$ as described in Equation 5 (see Figure 1). Given the rank reduction ratio $\Theta$, the rank of $\widehat{W}$ is determined by $r(\widehat{W}) = \text{round}((1 - \Theta) \cdot r(W))$. This selective low-rank approximation leverages the compact representation provided by SVD to enhance the model's ability to detect OOD instances effectively without requiring additional training. We demonstrate our method's ability to improve OOD detection (Table 1) while maintaining ID classification performance (Table 7) in Section 4.2 and Section 4.5.

**HyperParameter Search Algorithm**   Due to the presence of many weight matrices in CLIP, each consisting of hundreds of singular values, conducting a complete search over all combinations of low-rank approximation weight matrices is impractical. Therefore, we propose a greedy search algorithm to determine the rank reduction ratio for each weight matrix. Among all linear weight matrices in each encoder layer, we focus on $W_{\text{up}}$ as it is most effective according to our preliminary experiment. For simplicity, we assume both image and text encoders have $N$ encoder layers. As shown in Algorithm 1, we search by first enumerating all $N$ vision encoder layers sequentially from top to bottom and then all $N$ text encoder layers in the same way. This search order is concisely denoted as searching from $2N$ to the first layer in CLIP. We compare different search algorithms in Section 4.4. The rank reduction ratio for each layer is the objective in SeTAR which is searched among the candidate list $\Theta = \{\Theta_0, \Theta_1, \cdots, \Theta_J\}$ according to the loss on the validation set. We employ the LoCoOp loss (Equation 12) proposed in (Miyai et al., 2023a) as our loss

function. This loss requires only ID images. It contains an ID loss for ID image classification and an OOD loss to push away pseudo OOD features from the ID class text embeddings where the pseudo OOD features are from ID-irrelevant nuisances (Equation 10) (e.g., backgrounds) in CLIP's local features. We refer the readers to Miyai et al. (2023a) or Appendix B for more details.

For $\Theta_j \in \boldsymbol{\Theta}$, we remove $\Theta_j$ (in percent) singular values along with their corresponding singular vectors to obtain the approximated matrix $\widehat{W}_{\mathrm{up}}$ (Equation 5). It is worth noting that the rank reduction raio candidate list includes $\Theta_0 = 0$, indicating that the weight matrix has the chance to remain unmodified.

With the searched rank reduction ratio, the weight matrix $W_{\mathrm{up}}$ in each CLIP layer is replaced and updated with its approximation. The SeTAR can be easily applied to different ViT backbones (Table 8), by replacing the model weight matrices with their low-rank approximations in a similar approach. Then SeTAR detects the OOD data samples following MCM (Equation 1), GL-MCM (Equation 2) or other scoring-based OOD detection method with the approximated model. We provide an example procedure of the greedy search in Listing 1 for better understanding.

---

**Algorithm 1** The hyperparameter search in SeTAR.

**Data:** Valid set $D$.
**Input:** Layer length $2N$, rank reduction ratio candidates $\boldsymbol{\Theta}$ with length $J$, loss $\mathcal{L}_0$ on $D$ WITHOUT Se-TAR.
**Result:** Rank reduction ratio list $\mathbf{T}^*$ with length $2N$.
$\mathcal{L}^* = \mathcal{L}_0$ ;                    // Current best loss
**for** *LayerNum* $l \leftarrow 2N$ *to* 1 **do**
    $\widehat{W}^* = W_{\mathrm{up}}^l$  $T^*[l] = 0$
    **for** *counter* $j \leftarrow 1$ *to* $J$ **do**
        $r = \mathrm{round}((1 - \Theta[j]) \cdot \mathrm{rank}(W_{\mathrm{up}}^l))$
        $\widehat{W} = \sum_{i=1}^{r} \sigma_i^{\downarrow} u_i v_i^{\top}$
        Calclute loss $\mathcal{L}_j^l$ on $D$ by replacing $W_{\mathrm{up}}^l$ with $\widehat{W}$
        **if** $\mathcal{L}_j^l < \mathcal{L}^*$ **then**
            $\widehat{W}^* = \widehat{W}$; $T^*[l] = \Theta[j]$; $\mathcal{L}^* = \mathcal{L}_j^l$;
        **end**
    **end**
    $W_{\mathrm{up}}^l := \widehat{W}^*$
**end**
**return** $T^*$

---

## 3.2 OOD Detection with SeTAR-enhanced Low-rank Adaptation

**SeTAR** can be further combined with LoRA (Hu et al., 2022) as a novel low-rank adaptation method for OOD detection, which we refer to as **SeTAR+FT**. Specifically, we first apply SeTAR to the pre-trained CLIP model to obtain the reserved rank $r$ for each weight matrix $W$. Then we have

$$W = \widehat{W} + B \times A \tag{6}$$

$$B = \sum_{i=r+1}^{\min(m,n)} \sqrt{\sigma_i^{\downarrow}(W)} u_i \tag{7}$$

$$A = \sum_{i=r+1}^{\min(m,n)} \sqrt{\sigma_i^{\downarrow}(W)} v_i^{\top} \tag{8}$$

where $\widehat{W}$ is the low-rank approximation of $W$ found by SeTAR , with $A$ and $B$ being the minor singular components. During finetuning, we keep $\widehat{W}$ frozen and only update the low-rank matrix $A$ and $B$. In this way, we retain the principle singular components in the original weight matrix and only update the minor singular components.Unlike LoRA, which evenly distributes the finetuning rank budget across all layers, SeTAR+FT adjusts the rank for each layer, resulting in more effective and efficient fine-tuning (Table 2 and Figure 6). More details are provided in Section 4.3.

# 4 Experiments

## 4.1 Experimental Setup

**Dataset**   Following previous work (Ming et al., 2022; Miyai et al., 2023b), we use two real-world datasets created from ImageNet1K (Deng et al., 2009) and Pascal-VOC (Everingham et al., 2009) as the ID datasets. For OOD datasets, we follow Ming et al. (2022) to preprocess iNaturalist, SUN, Places and Texture, and follow Miyai et al. (2023b) to preprocess ImageNet22K and COCO data. For finetune experiments, we follow Miyai et al. (2023a) to use ImageNet1K as the ID dataset. The detailed description and statistics of the datasets are provided in Appendix C.

Table 1: **Training-free results of FPR95(FPR) and AUROC(AUC) compared to zero-shot baselines on CLIP-base. Bold** values represent the highest performance. † is cited from Miyai et al. (2023b), where ◇ represents the absence of reporting in the paper. * denotes the result of our re-run. − denotes the OOD dataset has overlapping categories with the ID dataset. We do not report standard deviations since no training is involved.

| Method | iNaturalist | | SUN | | Places | | Texture | | ImageNet22K | | COCO | | **Average** | |
|---|---|---|---|---|---|---|---|---|---|---|---|---|---|---|
| | FPR↓ | AUC↑ | FPR↓ | AUC↑ | FPR↓ | AUC↑ | FPR↓ | AUC↑ | FPR↓ | AUC↑ | FPR↓ | AUC↑ | FPR↓ | AUC↑ |
| **ImageNet1K** | | | | | | | | | | | | | | |
| **MCM Score** | | | | | | | | | | | | | | |
| Vanilla MCM† | 30.91 | 94.61 | 37.59 | 92.57 | 44.69 | 89.77 | 57.77 | 86.11 | - | - | - | - | 42.74 | 90.77 |
| Vanilla MCM* | 32.07 | 94.43 | 38.65 | 92.37 | 43.73 | 90.03 | 57.89 | 86.13 | - | - | - | - | 43.09 | 90.74 |
| SeTAR | **26.92** | **94.67** | **35.57** | **92.79** | **42.64** | **90.16** | **55.83** | **86.58** | - | - | - | - | **40.24** | **91.05** |
| **GL-MCM Score** | | | | | | | | | | | | | | |
| Vanilla GL-MCM† | 15.18 | 96.71 | 30.42 | 93.09 | 38.85 | 89.90 | 57.93 | 83.63 | - | - | - | - | 35.47 | 90.83 |
| Vanilla GL-MCM* | 15.34 | 96.62 | 30.65 | 93.01 | 37.76 | 90.07 | 57.41 | 83.73 | - | - | - | - | 35.29 | 90.86 |
| SeTAR | **13.36** | **96.92** | **28.17** | **93.36** | **36.80** | **90.40** | **54.17** | **84.59** | - | - | - | - | **33.12** | **91.32** |
| **Pascal-VOC** | | | | | | | | | | | | | | |
| **MCM Score** | | | | | | | | | | | | | | |
| Vanilla MCM† | 8.20 | 98.23 | 28.60 | 94.68 | ◇ | ◇ | 51.70 | 91.45 | 51.40 | 90.94 | 54.50 | 89.02 | 38.88 | 92.86 |
| Vanilla MCM* | 7.24 | 98.23 | 27.91 | 94.56 | 32.40 | 92.45 | 51.61 | 91.89 | 50.60 | 91.42 | 53.70 | 89.30 | 37.24 | 92.98 |
| SeTAR | **4.59** | **98.71** | **24.91** | **95.15** | **28.46** | **93.21** | **40.44** | **93.58** | **48.25** | **92.08** | **48.10** | **89.70** | **32.46** | **93.74** |
| **GL-MCM Score** | | | | | | | | | | | | | | |
| Vanilla GL-MCM† | 4.20 | 98.71 | 23.10 | 94.66 | ◇ | ◇ | 43.00 | 92.84 | 41.00 | 92.38 | 44.30 | 90.48 | 31.12 | 93.81 |
| Vanilla GL-MCM* | 4.33 | 98.81 | 22.94 | 94.63 | 26.20 | 93.11 | 41.61 | 92.88 | 37.88 | 93.17 | 43.70 | 90.71 | 29.44 | 93.88 |
| SeTAR | **3.66** | **98.96** | **21.93** | **94.81** | **25.04** | **93.62** | **20.35** | **96.36** | **31.47** | **94.31** | **40.70** | **91.19** | **23.86** | **94.87** |

**Settings**    Following existing studies (Ming et al., 2022; Miyai et al., 2023b,a), we use CLIP ViT-B/16[4] (Radford et al., 2021) as our backbone. Both image and text encoders have 12 layers. More results with different backbones are in Section 4.4. The rank reduction ratio candidates range from 0 to 40% in 5% intervals. We use a temperature of $1^5$, unless stated otherwise. In all experiments, we use one CLIP text prompt: "a photo of a [CLASS],", where [CLASS] is the ID class name. We set hyperparameters $\lambda$ (Equation 12) and top-K (Equation 10) according to the specific ID datasets and backbones. Detailed settings are in Table 12, with a sensitivity analysis in Section 4.4. For SeTAR+FT and LoRA experiments, the learning rate and epoch number are set to $1e-2$ and 5 for all experiments. The LoRA rank $r$ is set to match the trainable parameters of SeTAR+FT. Detailed settings are in Table 13. We report results from three runs with seeds 3, 4, 5[6]. All experiments are conducted on a single NVIDIA RTX 4090 GPU. The time cost for low-rank approximation with CLIP-base on the ImageNet1K validation set is about 20 minutes.

**Metrics**    We use the following metrics for evaluation. (1) the false positive rate (FPR95) for out-of-distribution (OOD) samples at a fixed true positive rate (TPR) of 95% for in-distribution samples, with lower values targeting better performance; and (2) the area under the receiver operating characteristic curve (AUROC) for OOD samples, with higher values indicating better performance.

**Baselines**    We evaluate SeTAR against MCM (Ming et al., 2022) and GL-MCM (Miyai et al., 2023b), state-of-the-art zero-shot OOD detection methods on CLIP. We also compare SeTAR+FT with fine-tuning baselines NPOS (Tao et al., 2023), CoOp (Zhou et al., 2022), LoCoOp (Miyai et al., 2023a), and LoRA (Hu et al., 2022). More details are in Appendix D.

### 4.2   Training-free Results

The training-free OOD detection performances are summarized in Table 1. Compared with zero-shot baselines, a salient observation is that on both MCM and GL-MCM, using SeTAR outperforms the vanilla method by a large margin across all OOD detection tasks. For example, using Pascal-VOC as ID, SeTAR yields a relatively average reduction of 12.84% FPR95 on MCM and 18.95% FPR95 on GL-MCM. Considering that SeTAR is generally applicable and training-free, these results are very encouraging. Comparing SeTAR with scoring function MCM and GL-MCM, SeTAR+GL-MCM performs better on all OOD detection tasks. However, the superiority of GL-MCM score over MCM appears to be contingent upon the choice of the model backbone. As evidenced in Table 8, SeTAR+MCM demonstrates superior performance with a relatively average FPR95 reduction of 8.30% compared to SeTAR+GL-MCM with CLIP-large as the backbone on ImageNet1K.

---

[4] `https://huggingface.co/openai/clip-vit-base-patch16`

[5] Temperature is set to 1.0 for the scaled CLIP logits, equivalent to the unscaled CLIP logits with a temperature of 100. We adopt the unscaled setting in our implementation.

[6] For SeTAR , the results are the same under different random seeds as it does not require training.

## 4.3 Fine-tuning Results

In this section, we compare SeTAR+FT with fine-tuning baselines, including NPOS (Tao et al., 2023), CoOp (Zhou et al., 2022), LoCoOp (Miyai et al., 2023a) and LoRA (Hu et al., 2022). LoCoOp is the state-of-the-art prompt-learning OOD detection method on CLIP. LoRA is a representative parameter-efficient fine-tuning method. Following previous work (Tao et al., 2023; Zhou et al., 2022; Miyai et al., 2023a), we report the results on the the ImageNet1K benchmark in Table 2. We observe that SeTAR+FT outperforms all baselines on both MCM and GL-MCM scoring functions. For example, with CLIP-base as the backbone, SeTAR+FT achieves a relatively average FPR95 reduction of 3.97% and 6.67% compared to LoCoOp and LoRA. Moreover, when scaled up to CLIP-large, SeTAR+FT outperforms LoCoOp and LoRA by relatively 17.92% and 12.45% FPR95 on the same benchmark. Similar results are observed on Swin Transformer (Liu et al., 2021), where SeTAR+FT outperforms LoRA by relatively 17.36% and 36.80% FPR95 on MSP and Energy scoring functions, respectively. The larger improvement on Swin Transformer may stem from its reliance on ImageNet training, making it prone to overfitting and weaker at OOD detection. Our method mitigates these issues, enhancing Swin's generalization to OOD instances. These results demonstrate the effectiveness and scalability of SeTAR+FT in improving the OOD detection performance.

Furthermore, as shown in Figure 6, SeTAR+FT demonstrates faster convergence and lower loss than LoRA, especially in OOD loss, indicating that SeTAR+FT is more effective in adapting the pre-trained weights to the OOD detection task.

Table 2: **Fine-tuning results on ImageNet1K benchmark. Bold** values indicate the highest performance. [†] is cited from Tao et al. (2023). [*] denotes our re-run results, $\pm$ indicates the standard deviation from 3 runs.

| **CLIP-base** | MCM Score | | GL-MCM Score | |
| --- | --- | --- | --- | --- |
| | FPR95↓ | AUROC↑ | FPR95↓ | AUROC↑ |
| NPOS[†] | 42.20 | 90.43 | 36.86 | 90.37 |
| CoOp[†] | 44.81 | 90.03 | 36.58 | 90.25 |
| LoCoOp[†] | 40.17 | 91.53 | 33.52 | 92.14 |
| LoCoOp[*] | $39.76_{\pm4.06}$ | $91.22_{\pm0.52}$ | $34.14_{\pm1.64}$ | $91.73_{\pm0.17}$ |
| LoRA[*] | $41.67_{\pm0.14}$ | $90.85_{\pm0.01}$ | $34.36_{\pm0.11}$ | $90.88_{\pm0.01}$ |
| SeTAR+FT | $\mathbf{38.77}_{\pm0.22}$ | $\mathbf{91.55}_{\pm0.01}$ | $\mathbf{32.19}_{\pm0.20}$ | $\mathbf{92.31}_{\pm0.05}$ |

| **CLIP-large** | MCM Score | | GL-MCM Score | |
| --- | --- | --- | --- | --- |
| | FPR95↓ | AUROC↑ | FPR95↓ | AUROC↑ |
| LoCoOp[*] | $40.74_{\pm3.80}$ | $91.13_{\pm0.79}$ | $46.74_{\pm4.19}$ | $89.32_{\pm0.80}$ |
| LoRA[*] | $38.62_{\pm0.07}$ | $91.66_{\pm0.02}$ | $43.39_{\pm0.01}$ | $89.76_{\pm0.03}$ |
| SeTAR+FT | $\mathbf{34.75}_{\pm0.55}$ | $\mathbf{92.86}_{\pm0.15}$ | $\mathbf{37.05}_{\pm0.59}$ | $\mathbf{91.83}_{\pm0.12}$ |

| **Swin-base** | MSP Score | | Energy Score | |
| --- | --- | --- | --- | --- |
| | FPR95↓ | AUROC↑ | FPR95↓ | AUROC↑ |
| LoRA[*] | $57.02_{\pm0.03}$ | $80.49_{\pm0.01}$ | $62.17_{\pm0.02}$ | $72.80_{\pm0.00}$ |
| SeTAR+FT | $\mathbf{47.12}_{\pm0.42}$ | $\mathbf{87.80}_{\pm0.44}$ | $\mathbf{39.29}_{\pm0.57}$ | $\mathbf{88.01}_{\pm0.51}$ |

## 4.4 Ablation Study

In this section, we conduct ablation studies with CLIP-base to understand our design choices.

**Image v.s. Text modality** Table 3 shows an ablation study on the modality involved in SeTAR. As shown, the vision modality outperforms the text modality, indicating the vision modality is more dominant in enhancing the model's performance. When considering the vision modality alone and the combined vision+text modality, the latter either outperforms or achieves comparable average results to the former. Consequently, we make modifications to both the vision and text modalities in SeTAR to enhance overall performance.

Table 3: **Ablation study on modality.**

| Score | Vision | | Text | | Vision+Text | |
| --- | --- | --- | --- | --- | --- | --- |
| | FPR↓ | AUC↑ | FPR↓ | AUC↑ | FPR↓ | AUC↑ |
| **ImageNet1K** | | | | | | |
| MCM | 40.27 | **91.24** | 42.78 | 90.50 | **40.24** | 91.05 |
| GL-MCM | 32.97 | **91.60** | 35.82 | 90.55 | 33.12 | 91.32 |
| **Pascal-VOC** | | | | | | |
| MCM | 33.19 | 93.45 | 33.47 | 93.42 | **32.46** | **93.74** |
| GL-MCM | 24.88 | 94.51 | 24.59 | 94.52 | **23.86** | **94.87** |

**Different Weight Types** In this part, we present empirical evidence for modifying $W_{up}$. We first compare the performance of SeTAR with different types of weight matrix in each Transformer layer, including $W_q$, $W_k$, $W_v$, $W_o$, $W_{up}$ and $W_{down}$. As shown in Figure 2 and Figure 3 of Appendx F, the $X$-axis denotes the number of weight matrixes (layers) that we have searched, while the $Y$-axis is the average AUROC and FPR95. The results show that $W_{up}$ consistently outperforms other weight matrices in terms of both AUROC and FPR95. In addition to weight matrics in each transformer layer, CLIP has one projection matrix $W_p$ on top of each encoder,

Table 4: **Comparison results of SeTAR with and without considering projection matrix** $W_p$.

| Score | Vanilla | | SeTAR w $W_p$ | | SeTAR w/o $W_p$ | |
| --- | --- | --- | --- | --- | --- | --- |
| | FPR↓ | AUC↑ | FPR↓ | AUC↑ | FPR↓ | AUC↑ |
| **ImageNet1K** | | | | | | |
| MCM | 43.09 | 90.74 | 41.79 | 90.74 | **40.24** | **91.05** |
| GL-MCM | 35.29 | 90.86 | 34.30 | 91.24 | **33.12** | **91.32** |
| **Pascal-VOC** | | | | | | |
| MCM | 37.24 | 92.98 | 35.94 | 93.32 | **32.46** | **93.74** |
| GL-MCM | 29.44 | 93.88 | **23.34** | 94.82 | 23.86 | **94.87** |

which serves to project image/text representations into a shared space. In Table 4, we compare the performance of SeTAR with and without modifying $W_p$. We search $W_p$ first right before searching the image/text encoder. The results show that frozen $W_p$ brings a relatively reduction of 4.20% FPR95. Consequently, we keep $W_p$ frozen in SeTAR.

**Different Search Algorithms** At each step of the greedy search, SeTAR traverses the subsequent $W_{up}$ in a predefined order and searches over different thresholds. We compare our method with two alternatives: modality-interleaved greedy search (MIS) and layer-exhaustive search (LES). MIS searches the image and text layers in an interleaved manner, while LES simultaneously searches over both layers and thresholds at each step. SeTAR-S, has linear complexity with respect to the number of model layers, similar to MIS, whereas LES has quadratic complexity. Table 5 presents the comparison results. SeTAR-S demonstrates better overall performance than MIS. Notably, MIS encounters limitations when the image and text towers have different layer counts (e.g., CLIP-large with 24 image layers and 12 text layers). Therefore, we choose SeTAR-S for better generalization. Compared to LES, SeTAR-S performs better in terms of both FPR95 and AUROC, as LES's locally optimal algorithm may not achieve a global optimal solution. These results validate the superiority of our top-to-bottom layer search strategy.

Table 5: **Results for different search algorithms.** Here LES, MIS and SeTAR-S stand for layer-exhaustive search, modality-interleave greedy search, and the search algorithm of SeTAR.

| Score | LES | | MIS | | SeTAR-S | |
|---|---|---|---|---|---|---|
| | FPR↓ | AUC↑ | FPR↓ | AUC↑ | FPR↓ | AUC↑ |
| **ImageNet1K** | | | | | | |
| MCM | 41.99 | 90.78 | 40.55 | 91.00 | **40.24** | **91.05** |
| GL-MCM | 33.90 | 91.08 | 33.36 | 91.29 | **33.12** | **91.32** |
| **Pascal-VOC** | | | | | | |
| MCM | 35.11 | 93.60 | 33.93 | 93.58 | **32.46** | **93.74** |
| GL-MCM | 24.48 | 94.57 | **22.87** | 94.84 | 23.86 | **94.87** |

**Different Prune Strategies** Inspired from SVD, SeTAR modify the model weights by pruning the minor singular components, and retains the principle components that contribute the most to the model's performance. To validate this design, we compare SeTAR with two alternatives: principal component pruning and random pruning pruning. Principal component takes the opposite approach, retaining minor components and pruning major ones. Random pruning, on the other hand, prunes weights randomly. As shown in Table 6, principle pruning suffers from a significant performance drop compared to SeTAR , while random pruning performs slightly better than principle pruning. These results demonstrate the effectiveness of SeTAR 's design choice in pruning the minor components.

Table 6: **Results for different pruning strategies.**

| Score | Principle | | Random | | Minor | |
|---|---|---|---|---|---|---|
| | FPR↓ | AUC↑ | FPR↓ | AUC↑ | FPR↓ | AUC↑ |
| **ImageNet1K** | | | | | | |
| MCM | 43.09 | 90.74 | 43.09 | 90.74 | **40.24** | **91.05** |
| GL-MCM | 35.29 | 90.86 | 35.29 | 90.86 | **33.12** | **91.32** |
| **Pascal-VOC** | | | | | | |
| MCM | 38.20 | 92.44 | 33.57 | 93.09 | **32.46** | **93.74** |
| GL-MCM | 25.36 | 93.67 | 26.20 | 94.66 | **23.86** | **94.87** |

**Sensitivity Analysis on $\lambda$ and top-K** In this section, we present the sensitivity analysis of the hyperparameters $\lambda$ (Figure 4) and top-K (Figure 5). As observed in Figure 4, the average AUROC remains stable at lower values and slightly decreases as $\lambda$ increases for both SeTAR+MCM and SeTAR+GL-MCM. Notably, the optimal setting of $\lambda$ may vary depending on the model backbone, with our experiments indicating that CLIP-large may require a larger $\lambda$ than CLIP-base. Despite this variation, the $\lambda$ parameter demonstrates strong transferability across datasets for the same backbone. Swapping the optimal $\lambda$ between ImageNet1K and Pascal-VOC has a minimal performance impact, consistently outperforming the vanilla method. With the VOC-optimized $\lambda$ on ImageNet1K, CLIP-base achieves an FPR95 of 40.91 and AUROC of 91.02, and CLIP-large reaches 46.73 FPR95 and 91.81 AUROC. Conversely, using the ImageNet1K-optimized $\lambda$ on Pascal-VOC, CLIP-base achieves 33.18 FPR95 and 93.65 AUROC, while CLIP-large attains 44.39 FPR95 and 92.3 AUROC.

Top-K controls the number of OOD regions considered in LoCoOp loss: higher values include more OOD regions, with top-K equal to the number of ID classes covering all OOD regions, and top-K set to 0 focusing solely on ID loss. The optimal top-K depends on the number of ID categories, making it non-transferable across datasets. However, SeTAR remains robust to top-K variations, as shown in Figure 5, except at extreme values (0 or the maximum number of classes). We recommend setting top-K to around 30% of the total categories, such as 300 for ImageNet1K and 4 for Pascal-VOC. For the Swin-base model, top-K at 300 on ImageNet1K yields an FPR95 of 56.82 and AUROC of 85.68 with MSP, and an FPR95 of 52.56 and AUROC of 84.51 with Energy.

## 4.5 Analyses

**Can SeTAR Improve Image Classification?**
To evaluate the impact of SeTAR and SeTAR+FT on classification accuracy, we present our results on ID dataset ImageNet1K and OOD datasets SUN, Places and Texture in Table 7[7]. SeTAR effectively maintains the average accuracy, with minor variations observed across different datasets. Among the fine-tuned baselines, LoCoOp exhibits a 1% decrease in accuracy compared to Vanilla CLIP, whereas LoRA

Table 7: **Image classification results with different methods.** We use ImageNet1K (IN1K) as ID dataset. * denotes the results of our re-run. The results are averaged over 3 runs.

| Method | IN1K | SUN | Places | Texture | **Average** |
|---|---|---|---|---|---|
| Vanilla CLIP* | 64.07 | 75.77 | 45.65 | 43.60 | 57.27 |
| LoCoOp* | 64.93 | 75.89 | 46.47 | 37.79 | 56.27 |
| LoRA* | 65.43 | 76.86 | 46.58 | **43.98** | 58.21 |
| SeTAR | 63.97 | 75.50 | 45.81 | 43.76 | 57.26 |
| SeTAR+FT | **67.02** | **77.94** | **46.64** | 43.28 | **58.72** |

shows an improvement of 0.94%. Notably, SeTAR+FT surpasses both baselines, improving the average accuracy by 1.45% compared to Vanilla CLIP. These results highlight the efficacy of SeTAR and SeTAR+FT in improving OOD detection without compromising classification accuracy.

**SeTAR is Effective on Different Architectures and Score Functions** We expand on Table 1 with results on ViT and CNN backbones and various score functions. For ViT-based models, we evaluate OOD detection using CLIP-large[8] and Swin Transformer[9] (Liu et al., 2021), alongside CLIP-base. The Swin Transformer (Liu et al., 2022) is trained on ImageNet1K. Since it lacks a text encoder, we apply SeTAR to the image ViT only. For Swin Transformer, we use two common scoring functions: MSP (Hendrycks & Gimpel, 2017), which leverages softmax confidence, and the Energy score (Liu et al., 2020), with $T = 0.1$ for OOD detection. We also integrate CLIP-base with the NegLabel score function (Jiang et al., 2024), which uses large-scale negative labels. As shown in Table 8, SeTAR consistently outperforms baselines across all backbones and scoring functions, significantly reducing FPR95 by relatively 20.61% with the Energy score on Swin Transformer. These results demonstrate SeTAR 's effectiveness in improving OOD detection for unimodal image encoders, with further confirmation from SeTAR+FT results (Table 2) across different model backbones.

Table 8: **Results for different ViT backbones.**

| Backbone | Score | Vanilla Method | | SeTAR | |
|---|---|---|---|---|---|
| | | FPR↓ | AUC↑ | FPR↓ | AUC↑ |
| **ImageNet1K** | | | | | |
| CLIP-base | NegLabel | 25.40 | 94.21 | **23.09** | **94.48** |
| CLIP-large | MCM | 37.19 | 91.73 | **36.26** | **91.92** |
| CLIP-large | GL-MCM | 40.65 | 89.98 | **39.54** | **90.22** |
| Swin-base | MSP | 59.25 | 84.12 | **56.05** | **85.77** |
| Swin-base | Energy | 65.01 | 76.10 | **51.61** | **84.42** |
| **Pascal-VOC** | | | | | |
| CLIP-large | MCM | 52.21 | 91.68 | **42.57** | **92.91** |
| CLIP-large | GL-MCM | 43.96 | 92.45 | **31.12** | **94.00** |

We further explore SeTAR's potential on CNN architecture, and compare it with methods such as Softmax, Energy (Wu et al., 2023), ReAct (Sun et al., 2021), DICE (Sun & Li, 2022), and ASH (Djurisic et al., 2023) on ResNet50[10]. Since ResNet lacks local features for OOD loss, we conduct experiments using only ID loss. We apply low-rank approximation to the in- and out-feature dimensions of the convolutional layers,

Table 9: **Results on ResNet50**. We use ImageNet1K as the ID dataset. [†] is cited from Djurisic et al. (2023).

| Method | FPR↓ | AUC↑ | Method | FPR↓ | AUC↑ |
|---|---|---|---|---|---|
| Softmax[†] | 66.95 | 81.99 | ASH-P[†] | 50.32 | 89.04 |
| Energy[†] | 58.41 | 86.17 | ASH-B[†] | 22.73 | 95.06 |
| ReAct[†] | 31.43 | 92.95 | ASH-S[†] | 22.80 | 95.12 |
| DICE[†] | 34.75 | 90.77 | SeTAR | **22.38** | **95.25** |

combined with ASH for search. As shown in Table 9, SeTAR establishes new state-of-the-art results on ResNet, demonstrating its effectiveness across both ViT and CNN architectures.

**Near-OOD Results** To further evaluate SeTAR's performance on diverse OOD tasks, we test it on a more challenging near-OOD setting using ImageNet1K as the ID dataset and SSB-Hard (Vaze et al., 2022) as the OOD dataset. As shown in Table 10, SeTAR and SeTAR+FT outperform the baselines, demonstrating superior performance in near-OOD scenarios.

Table 10: **Near-OOD results on CLIP-base**.

| Method | Category | MCM Score | | GL-MCM Score | |
|---|---|---|---|---|---|
| | | FPR↓ | AUC↑ | FPR↓ | AUC↑ |
| Vanilla | Training-Free | 89.28 | 63.88 | 85.62 | 67.63 |
| SeTAR | Training-Free | 88.29 | 64.20 | **84.03** | 68.29 |
| LoCoOp | Training-Free | 89.72 | 63.45 | 86.79 | 65.93 |
| LoRA | Finetuning | 88.52 | 65.38 | 84.39 | 68.85 |
| SeTAR+FT | Finetuning | **87.16** | **68.13** | 84.72 | **70.42** |

---

[7]We do not report classification accuracy on iNaturalist as we failed to match the labels for the OOD test set.

[8]https://huggingface.co/openai/clip-vit-large-patch14

[9]https://huggingface.co/microsoft/swinv2-base-patch4-window16-256

[10]https://download.pytorch.org/models/resnet50-19c8e357.pth

# 5    Related Work

**Out-of-Distribution Detection**   Previous work explores OOD detection with unimodal (DeVries & Taylor, 2018; Hendrycks & Gimpel, 2017; Hu & Khan, 2021; Zheng et al., 2020; Zhou et al., 2021) and multimodal (Fort et al., 2021; Ming et al., 2022; Tao et al., 2023; Miyai et al., 2023a) models.  Numerous methodologies (Lee et al., 2018; Huang et al., 2021; Sun et al., 2022; Wang et al., 2022; Wu et al., 2023) have been developed to tackle OOD detection in computer vision. Existing CLIP-based OOD detection methods include zero-shot (Fort et al., 2021; Ming et al., 2022; Miyai et al., 2023b; Dai et al., 2023; Wang et al., 2023; Jiang et al., 2024) and fine-tuning (Ming & Li, 2023; Tao et al., 2023; Miyai et al., 2023a). Zero-shot methods like MCM (Ming et al., 2022) and GL-MCM (Miyai et al., 2023b) don't require in-distribution training data but may perform suboptimally due to domain gaps. Other approaches integrate external knowledge. For example, CLIPN (Wang et al., 2023) pre-trains a novel NO-encoder on the CC-3M dataset (Sharma et al., 2018) to empower CLIP's "no" logic for zero-shot evaluation. NegLabel (Jiang et al., 2024) demonstrates better performance than CLIPN by introducing large-scale negative labels for enhanced label scoring. Fine-tuning methods (Ming & Li, 2023; Tao et al., 2023; Miyai et al., 2023a) improve OOD detection by adapting to in-distribution data but risk damaging the pretraining representations, needing careful training strategies. CNN-based OOD detection methods, including ReAct (Sun et al., 2021), ASH (Djurisic et al., 2023), DICE (Sun & Li, 2022), CIDER (Ming et al., 2023), PALM (Lu et al., 2024), and Hopfield Boosting (Hofmann et al., 2024), have also demonstrated strong results.  However, methods like ReAct and ASH rely on the assumption that ID and OOD images produce distinct activations in models trained on ID data. This assumption does not hold in large-scale pre-trained models like CLIP, where activations for ID and OOD images are not significantly different, limiting the effectiveness of such approaches in enhancing CLIP's zero-shot OOD detection capabilities. SeTAR, in contrast, offers high compatibility with various scoring functions (e.g., MCM, GL-MCM, MSP, Energy), multiple model backbones (e.g., CLIP, Swin, ResNet), and advanced OOD techniques such as NegLabel. Designed to be both lightweight and efficient, SeTAR addresses the demand for resource-efficient solutions in OOD detection.

**Low-rank Approximations of Weight Matrices**   Neural networks trained with over-parameterization often exhibit low-rank properties (Oymak et al., 2019).  These properties are utilized in both model training (Povey et al., 2018; Hu et al., 2022) and post-hoc processing (Hajimolahoseini et al., 2021; Sharma et al., 2023). In training, some works (Sainath et al., 2013; Zhang et al., 2014; Zhao et al., 2016) impose low-rank constraints, while LoRA (Hu et al., 2022) adapts pretrained LLMs to downstream tasks using trainable low-rank matrices. For post-hoc processing, pruning methods (Yu et al., 2017; Hajimolahoseini et al., 2021) reduce weight matrix ranks by retaining top-K components from SVD. While pruning preserves model behavior, performance declines with increased intervention. LASER (Sharma et al., 2023) focuses on pruning individual layers to enhance factual answering capabilities. It utilizes a simple greedy search strategy on a validation set, which is not applicable for OOD detection due to the absence of a validation set. In contrast, our approach introduces a selective rank reduction strategy specifically tailored for OOD detection. We systematically analyze and compare different greedy search techniques, evaluating their effectiveness across various layers and model backbones.

# 6    Conclusion

We propose SeTAR , a simple and effective OOD detection method using post-hoc low-rank approximation on weight matrices $W_{up}$ with a top-down, image-to-text greedy search. SeTAR offers several advantages: (1) training-free, (2) scalable to unimodal and multimodal models, and (3) complementary to existing OOD scoring functions. Building on SeTAR , we introduce SeTAR-FT, a finetuning method that adapts the model to in-distribution data for improved OOD detection. We evaluate SeTAR and SeTAR-FT on large-scale benchmarks, including ImageNet1K and Pascal-VOC. Results show that both achieve state-of-the-art OOD detection performance.  We hope our work inspires further research and contributes to more robust and reliable models.

## Acknowledgements

This project was supported by National Natural Science Foundation of China (No. 62306132, No. 62106138). We thank the anonymous reviewers for their insightful feedbacks on this work.

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

# A    Impact Statements

**Limitation**    While we demonstrate the effectiveness of our method on OOD detection, we acknowledge that our work has several limitations. First, despite we show the robustness of our method to hyperparameters, the optimal hyperparameters may vary across different model backbones. Future work is needed to explore the autonomous selection of hyperparameters. Second, we design SeTAR+FT in a simple and straightforward manner, which may not be the most efficient or effective way to adapt the model to the ID downstream data. More sophisticated strategies for model adaptation are worth exploring in future research. Third, we only conduct experiments to detect visual OOD inputs and ignore inputs in other modalities such as textual, audio and video. This is primarily because our model is based on CLIP. Exploring the development of OOD detectors across diverse modalities remains an active research topic for future investigation.

**Ethical Considerations**    Our study addresses the challenge of OOD detection through low-rank approximation, which is particularly relevant for ensuring the reliability and trustworthiness of vision-and-language pre-trained models. Future investigations on fairness, privacy and transparency neural-based models should be encouraged to mitigate the existing data biases and safety problems for a responsible, helpful and trustworthy AI system in diverse real-world applications.

**Future Societal Consequences**    Our proposed SeTAR achieves impressive OOD detection performance, which is beneficial to various real-world machine learning applications, such as healthcare and autonomous vehicles. The identification of anomalies or unexpected data points is crucial for decision-making and risk management with AI models. A better OOD detector facilitates the development of trustworthy machine-learning models that can reject unknown data inputs and help alleviate the hallucination problem. Moreover, better OOD detectors like SeTAR can help to select and label the unfamiliar data samples to further train a stronger model in the wild.

# B    Loss Function

To improve the model's OOD detection ability, it is crucial to define a loss function that pushes OOD samples far from ID samples while keeping ID samples close to each other. However, since OOD samples are unavailable during development, we address this issue by using the LoCoOp loss (Miyai et al., 2023a) for both SeTAR and SeTAR+FT. The main idea is to create pseudo OOD features with ID-irrelevant nuisances (e.g., backgrounds) in CLIP's local features.

Specifically, we divide the image into patches, represented by the set of all patch indices $I = \{0, 1, 2, \ldots, H \times W - 1\}$, where $H$ and $W$ denote the height and width of the patch features. Next, we compute the cosine similarity between the image patch features $p_i^v$ and the text features $h_c^t$ of the image label. The classification prediction probabilities for each patch $i$ are then given by:

$$p_i(y = m|\mathbf{x}) = \frac{\exp(\text{cos\_sim}(p_i^v, h_c^t)/\tau')}{\sum_{c=1}^{K} \exp(\text{cos\_sim}(p_i^v, h_c^t)/\tau')} \tag{9}$$

For a given image patch related to an ID category, the corresponding ID label should be among its top-K predictions. Conversely, for patches unrelated to the ID label, such as background regions, the ID label should be excluded from the top-K predictions. Based on this intuition, the indices of ID-irrelevant regions within an image are defined by Equation 10, where $\text{rank}(p_i(y = \mathbf{y}|\mathbf{x}))$ denotes the rank of the true class $\mathbf{y}$ among all ID classes, and K is the hyperparameter.

$$J = \{i \mid \text{rank}(p_i(y = \mathbf{y}|\mathbf{x})) > \text{K}\} \tag{10}$$

After identifying out-of-distribution (OOD) regions, it is expected that their image features will differ significantly from the ID text embeddings. To enhance this distinction, entropy maximization is employed to increase the entropy of $p_j(y|\mathbf{x})$, where $p_j$ denotes the classification prediction probabilities for region $j \in J$. The entropy maximization is formally defined as follows:

$$\mathcal{L}_{\text{ood}} = -H(p_j) \tag{11}$$

Here, $H(\cdot)$ represents the entropy function. The overall loss function combines the ID loss (cross-entropy loss for ID predictions) with the OOD loss. Here $\lambda$ is the hyperparameter that regulates the proportion of the OOD loss.

$$\mathcal{L} = \mathcal{L}_{\text{id}} + \lambda \mathcal{L}_{\text{ood}} \tag{12}$$

## C Data

Table 11: **The statistics of the dataset used in this paper.** 'ID' and 'OOD' denote in-distribution and out-of-distribution, respectively.

| Data | Type | Valid Size | Test Size |
|---|---|---|---|
| ImageNet1K (Deng et al., 2009) | ID | 1,000 | 50,000 |
| Pascal-VOC (Everingham et al., 2009) | ID | 94 | 906 |
| iNaturalist (Van Horn et al., 2018) | OOD | 0 | 10,000 |
| SUN (Xiao et al., 2010) | OOD | 0 | 10,000 |
| Places (Zhou et al., 2017) | OOD | 0 | 10,000 |
| Texture (Cimpoi et al., 2014) | OOD | 0 | 5,640 |
| ImageNet22K (Russakovsky et al., 2015) | OOD | 0 | 18,335 |
| COCO (Lin et al., 2014) | OOD | 0 | 1,000 |

We use two real-world datasets created from ImageNet1K (Deng et al., 2009) and Pascal-VOC (Everingham et al., 2009) as the ID dataset. We use ImageNet-1K validation set as the ID test set following Ming et al. (2022), and preprocess Pascal-VOC following Miyai et al. (2023b). we build two ID validation sets for low-rank approximation. The ID validation set of ImageNet1K is collected by sampling one image for each label from the ImageNet1K training set. For Pascal-VOC, For Pascal-VOC, We randomly sample 10% images as the ID validation set and leave the rest as the ID test set.

For OOD datasets, we follow Ming et al. (2022) to preprocess iNaturalist, SUN, Places and Texture, and follow Miyai et al. (2023b) to preprocess ImageNet22K and COCO data. We only evaluate the OOD datasets that have no overlapping categories as the ID dataset.

We provide more details about the datasets used in our experiments, in terms of data sources, preprocessing, and the statistics for each dataset, as shown in Table 11 and below.

**ImageNet1K** We use the ImageNet-1000 (ILSVRC2012) (Deng et al., 2009) dataset for ID validation and testing. The original dataset contains 1.2 million training images and 50,000 validation images from 1000 classes, and is widely used for image classification. We follow Ming et al. (2022) to construct the ImageNet1K ID test set from the validation set. Additionally, we curate an ImageNet1K ID validation set from the training set by randomly selecting one image for each label.

**Pascal-VOC** The Pascal VOC (Visual Object Classes) (Everingham et al., 2009) dataset is a benchmark dataset widely used in computer vision, featuring annotated images across multiple object categories. We use the Pascal-VOC subset collected by Miyai et al. (2023b) as the ID dataset, each image has single-class ID objects and one or more OOD objects. The ID validation and test set are split by 1:9 for each class, resulting in 94 and 906 images, respectively.

**iNaturalist** iNaturalist (Van Horn et al., 2018) is a biodiversity dataset containing millions of labeled images of plants, animals, and insects. Ming et al. (2022) construct a subset with 10,000 images by de-duplicating concepts overlapped with ID datasets.

**Places** Places (Zhou et al., 2017) is a scene-centric database with 205 scene categories and 2.5 million images. We use the SUN subset collected by Ming et al. (2022) as the OOD test set, which contains 10,000 images that are not overlapped with the ID classes.

**SUN** SUN (Scene UNderstanding) (Xiao et al., 2010) is a comprehensive collection of labeled images representing a diverse range of indoor and outdoor scenes. We use the SUN subset collected by Ming et al. (2022) as the OOD test set, which contains 10,000 images that are not overlapped with the ID classes.

**Texture** The Texture dataset (DTD) (Cimpoi et al., 2014) comprises 5640 images categorized into 47 terms inspired by human perception, aimed at replicating human-like texture recognition in machines. Again, we use the subset collected by Ming et al. (2022) as the OOD test set.

**ImageNet22K** The ImageNet-22K dataset (Russakovsky et al., 2015), formerly known as ImageNet-21K, addresses the underestimation of its additional value compared to the standard ImageNet-1K

pretraining, aiming to provide high-quality pretraining for a broader range of models. We use the filtered subset collected by Wang et al. (2021) as the OOD test set for MC-COCO and Pascal-VOC ID test sets.

**COCO**   Miyai et al. (2023b) curated an MS-COCO OOD test set (COCO for short) with 1,000 images that are not overlapped with the Pascal-VOC ID classes, which we use as OOD testing data for Pascal-VOC ID test set.

## D   Fine-tune Baselines

We compare SeTAR+FT with 4 finetuning-based baselines. These baselines include:

- **NPOS.** NPOS (Tao et al., 2023) generates virtual anomalies in low-probability regions of ID data without relying on distribution assumptions, enhancing discrimination during training.

- **CoOp.** CoOp (Zhou et al., 2022) optimizes prompts for vision-language models with learnable context vectors for efficient few-shot learning.

- **LoCoOp.** LoCoOp (Miyai et al., 2023a) improves upon CoOp by leveraging CLIP's local features to better distinguish between ID and OOD samples, achieving higher detection accuracy with less training data. We follow the official code[11] to prepare and fine-tune the LoCoOp with CLIP-base and CLIP-large. Follow Miyai et al. (2023a), the top-K, $\lambda$, learning rate and epoch num are set to 200, 0.25, 0.002 and 50. Temperature is set to 1 and the text prompt is initiated with "`X X X X X X X X X X X X X X X X [CLASS]`", where [CLASS] is the ID class name. We average the results from 3 seeds finetuned with 1-shot ImageNet1K valid data.

- **LoRA.** LoRA (Hu et al., 2022) is a low-rank adaptation method that injects trainable low-rank decomposition matrices into the pre-trained model to adapt to downstream tasks. We apply low-rank adaptation to the same weight type as SeTAR+FT, the rank of each layer is set to match the trainable parameters of SeTAR. Details settings can be found in Table 13.

## E   HyperParameters Settings

The hyperparameters for SeTAR are shown in Table 12. And the hyperparameters for SeTAR+FT and LoRA are shown in Table 13.

Table 12: **Hyperparameters for SeTAR .** Temperature is set to 1 except for Swin-base with Energy score, where it is set to 0.1.

| Backbone | Dataset | $\lambda$ | top-K |
|---|---|---|---|
| CLIP-base | ImageNet1K | 0.10 | 300 |
| | Pascal-VOC | 0.05 | 4 |
| CLIP-large | ImageNet1K | 0.50 | 300 |
| | Pascal-VOC | 0.30 | 6 |
| Swin-base | ImageNet1K | 0.01 | 700 |

Table 13: **Hyperparameters for SeTAR+FT and LoRA on ImageNet1K.** Temperature is set to 1 except for Swin-base with Energy score, which is set to 0.1.

| Backbone | $\lambda$ | top-K | LR | Epoch | Rank for LoRA | Alpha for LoRA |
|---|---|---|---|---|---|---|
| CLIP-base | 0.10 | 300 | 0.01 | 5 | 32 | 16 |
| CLIP-large | 0.50 | 300 | 0.01 | 5 | 64 | 16 |
| Swin-base | 0.01 | 700 | 0.01 | 5 | 112 | 16 |

---

[11]`https://github.com/AtsuMiyai/LoCoOp`

# F   More Detailed Experiment Results

In this section, we present additional detailed results from the main paper. This includes the detailed results of fine-tuned baselines on the ImageNet1K benchmark in Table 14; detailed ablation results on modality, $W_p$, $\lambda$, and top-K in Table 15, Table 16, Table 19, and Table 21; and detailed results of SeTAR with different search algorithms, prune strategies and backbones in Table 18, Table 20, Table 17 and Table 22.

Table 14: **Detail results of FPR95(FPR) and AUROC(AUC) compared with fine-tuned baselines on ImageNet1K benchmark.** [†] is cited from Tao et al. (2023). [*] denotes the results of our re-run.

| Method | iNaturalist | | SUN | | Places | | Texture | | **Average** | |
|---|---|---|---|---|---|---|---|---|---|---|
| | FPR↓ | AUC↑ | FPR↓ | AUC↑ | FPR↓ | AUC↑ | FPR↓ | AUC↑ | FPR↓ | AUC↑ |
| **CLIP-base** | | | | | | | | | | |
| **MCM Score** | | | | | | | | | | |
| NPOS[†] | 19.59 | 95.68 | 48.26 | 89.70 | 49.82 | 88.77 | 51.12 | 87.58 | 42.20 | 90.43 |
| CoOp[†] | 43.38 | 91.26 | 38.53 | 91.95 | 46.68 | 89.09 | 50.64 | 87.83 | 44.81 | 90.03 |
| LoCoOp[†] | 38.49 | 92.49 | 33.27 | 93.67 | 39.23 | 91.07 | 49.25 | 89.13 | 40.17 | 91.53 |
| LoCoOp[*] | 31.33 | 93.64 | 33.68 | 93.37 | 42.31 | 90.10 | 51.72 | 87.75 | 39.76 | 91.22 |
| LoRA[*] | 30.50 | 94.51 | 35.08 | 92.87 | 43.20 | 90.03 | 57.91 | 85.97 | 41.67 | 90.85 |
| SeTAR+FT | 32.95 | 93.41 | 30.26 | 93.81 | 38.56 | 91.24 | 53.32 | 87.72 | 38.77 | 91.55 |
| **GL-MCM Score** | | | | | | | | | | |
| NPOS[†] | 18.70 | 95.36 | 38.99 | 90.33 | 41.86 | 89.36 | 47.89 | 86.44 | 36.86 | 90.37 |
| CoOp[†] | 21.30 | 95.27 | 31.66 | 92.16 | 40.44 | 89.31 | 52.93 | 84.25 | 36.58 | 90.25 |
| LoCoOp[†] | 24.61 | 94.89 | 25.62 | 94.59 | 34.00 | 92.12 | 49.86 | 87.49 | 33.52 | 92.14 |
| LoCoOp[*] | 18.97 | 95.90 | 27.33 | 94.31 | 37.29 | 90.75 | 52.98 | 85.95 | 34.14 | 91.73 |
| LoRA[*] | 15.16 | 96.48 | 27.99 | 93.48 | 36.74 | 90.30 | 57.56 | 83.24 | 34.36 | 90.88 |
| SeTAR+FT | 21.62 | 95.43 | 23.38 | 94.89 | 32.60 | 91.93 | 51.18 | 87.01 | 32.19 | 92.31 |
| **CLIP-large** | | | | | | | | | | |
| **MCM Score** | | | | | | | | | | |
| LoCoOp[*] | 41.84 | 91.77 | 35.28 | 92.78 | 41.52 | 90.01 | 44.33 | 89.96 | 40.74 | 91.13 |
| LoRA[*] | 34.65 | 93.65 | 29.78 | 94.21 | 36.65 | 91.59 | 53.40 | 87.18 | 38.62 | 91.66 |
| SeTAR+FT | 22.41 | 95.83 | 40.07 | 91.98 | 45.19 | 90.13 | 31.37 | 93.48 | 34.75 | 92.86 |
| **GL-MCM Score** | | | | | | | | | | |
| LoCoOp[*] | 51.56 | 89.45 | 37.85 | 92.43 | 43.86 | 89.33 | 53.72 | 86.05 | 46.74 | 89.32 |
| LoRA[*] | 41.00 | 91.96 | 31.69 | 93.85 | 39.65 | 90.79 | 61.22 | 82.46 | 43.39 | 89.76 |
| SeTAR+FT | 36.56 | 91.93 | 34.81 | 93.08 | 41.08 | 90.66 | 35.74 | 91.66 | 37.05 | 91.83 |
| **Swin-base** | | | | | | | | | | |
| **MSP Score** | | | | | | | | | | |
| LoRA[*] | 43.14 | 87.02 | 62.66 | 78.04 | 67.95 | 74.90 | 54.34 | 81.99 | 57.02 | 80.49 |
| SeTAR+FT | 29.10 | 94.38 | 52.39 | 86.75 | 57.67 | 85.80 | 49.31 | 84.28 | 47.12 | 87.80 |
| **Energy Score** | | | | | | | | | | |
| LoRA[*] | 62.49 | 71.48 | 65.05 | 71.47 | 75.00 | 63.24 | 46.13 | 85.02 | 62.17 | 72.80 |
| SeTAR+FT | 29.76 | 91.56 | 42.76 | 87.06 | 51.73 | 82.85 | 32.90 | 90.56 | 39.29 | 88.01 |

Table 15: **Detail results of ablation study on modality.** We use CLIP-B/16 as a backbone.

| Method | iNaturalist | | SUN | | Places | | Texture | | ImageNet22K | | COCO | | Average | |
|---|---|---|---|---|---|---|---|---|---|---|---|---|---|---|
| | FPR↓ | AUC↑ | FPR↓ | AUC↑ | FPR↓ | AUC↑ | FPR↓ | AUC↑ | FPR↓ | AUC↑ | FPR↓ | AUC↑ | FPR↓ | AUC↑ |
| **ImageNet1K** | | | | | | | | | | | | | | |
| **MCM Score** | | | | | | | | | | | | | | |
| Visual | 29.69 | 94.58 | 35.15 | 92.99 | 41.25 | 90.45 | 55.00 | 86.92 | - | - | - | - | 40.27 | 91.24 |
| Text | 30.21 | 94.33 | 38.39 | 92.27 | 44.48 | 89.74 | 58.05 | 85.64 | - | - | - | - | 42.78 | 90.50 |
| Visual+Text | 26.92 | 94.67 | 35.57 | 92.79 | 42.64 | 90.16 | 55.83 | 86.58 | - | - | - | - | 40.24 | 91.05 |
| **GL-MCM Score** | | | | | | | | | | | | | | |
| Visual | 13.81 | 96.93 | 27.89 | 93.67 | 36.12 | 90.74 | 54.06 | 85.06 | - | - | - | - | 32.97 | 91.60 |
| Text | 15.44 | 96.54 | 30.77 | 92.78 | 38.95 | 89.71 | 58.14 | 83.17 | - | - | - | - | 35.82 | 90.55 |
| Visual+Text | 13.36 | 96.92 | 28.17 | 93.36 | 36.80 | 90.40 | 54.17 | 84.59 | - | - | - | - | 33.12 | 91.32 |
| **Pascal-VOC** | | | | | | | | | | | | | | |
| **MCM Score** | | | | | | | | | | | | | | |
| Visual | 4.13 | 98.63 | 26.31 | 94.58 | 30.44 | 92.58 | 42.48 | 93.20 | 45.19 | 92.36 | 50.60 | 89.36 | 33.19 | 93.45 |
| Text | 7.29 | 98.06 | 26.33 | 94.68 | 30.25 | 92.65 | 44.57 | 92.25 | 44.38 | 92.40 | 48.00 | 90.45 | 33.47 | 93.42 |
| Visual+Text | 4.59 | 98.71 | 24.91 | 95.15 | 28.46 | 93.21 | 40.44 | 93.58 | 48.25 | 92.08 | 48.10 | 89.70 | 32.46 | 93.74 |
| **GL-MCM Score** | | | | | | | | | | | | | | |
| Visual | 3.90 | 98.89 | 22.40 | 94.27 | 26.22 | 93.03 | 22.87 | 95.97 | 31.40 | 94.10 | 42.50 | 90.81 | 24.88 | 94.51 |
| Text | 3.55 | 99.01 | 21.26 | 94.48 | 24.87 | 92.96 | 30.89 | 94.07 | 29.86 | 94.49 | 37.10 | 92.09 | 24.59 | 94.52 |
| Visual+Text | 3.66 | 98.96 | 21.93 | 94.81 | 25.04 | 93.62 | 20.35 | 96.36 | 31.47 | 94.31 | 40.70 | 91.19 | 23.86 | 94.87 |

Table 16: **Detail results of SeTAR with and without considering projection matrix $W_p$.** We use CLIP-B/16 as a backbone.

| Method | iNaturalist | | SUN | | Places | | Texture | | ImageNet22K | | COCO | | Average | |
|---|---|---|---|---|---|---|---|---|---|---|---|---|---|---|
| | FPR↓ | AUC↑ | FPR↓ | AUC↑ | FPR↓ | AUC↑ | FPR↓ | AUC↑ | FPR↓ | AUC↑ | FPR↓ | AUC↑ | FPR↓ | AUC↑ |
| **ImageNet1K** | | | | | | | | | | | | | | |
| **MCM Score** | | | | | | | | | | | | | | |
| Vanilla MCM | 32.07 | 94.43 | 38.65 | 92.37 | 43.73 | 90.03 | 57.89 | 86.13 | - | - | - | - | 43.09 | 90.74 |
| SeTAR w $W_p$ | 35.21 | 93.06 | 33.50 | 93.16 | 41.02 | 90.50 | 57.41 | 86.22 | - | - | - | - | 41.79 | 90.74 |
| SeTAR w/o $W_p$ | 26.92 | 94.67 | 35.57 | 92.79 | 42.64 | 90.16 | 55.83 | 86.58 | - | - | - | - | 40.24 | 91.05 |
| **GL-MCM Score** | | | | | | | | | | | | | | |
| Vanilla GL-MCM | 15.34 | 96.62 | 30.65 | 93.01 | 37.76 | 90.07 | 57.41 | 83.73 | - | - | - | - | 35.29 | 90.86 |
| SeTAR w $W_p$ | 19.08 | 95.69 | 26.52 | 93.93 | 35.18 | 91.01 | 56.42 | 84.34 | - | - | - | - | 34.30 | 91.24 |
| SeTAR w/o $W_p$ | 13.36 | 96.92 | 28.17 | 93.36 | 36.80 | 90.40 | 54.17 | 84.59 | - | - | - | - | 33.12 | 91.32 |
| **Pascal-VOC** | | | | | | | | | | | | | | |
| **MCM Score** | | | | | | | | | | | | | | |
| Vanilla MCM | 7.24 | 98.23 | 27.91 | 94.56 | 32.40 | 92.45 | 51.61 | 91.89 | 50.60 | 91.42 | 53.70 | 89.30 | 37.24 | 92.98 |
| SeTAR w $W_p$ | 6.54 | 98.40 | 26.95 | 94.88 | 30.61 | 92.91 | 49.40 | 92.09 | 51.16 | 91.84 | 51.00 | 89.83 | 35.94 | 93.32 |
| SeTAR w/o $W_p$ | 4.59 | 98.71 | 24.91 | 95.15 | 28.46 | 93.21 | 40.44 | 93.58 | 48.25 | 92.08 | 48.10 | 89.70 | 32.46 | 93.74 |
| **GL-MCM Score** | | | | | | | | | | | | | | |
| Vanilla GL-MCM | 4.33 | 98.81 | 22.94 | 94.63 | 26.20 | 93.11 | 41.61 | 92.88 | 37.88 | 93.17 | 43.70 | 90.71 | 29.44 | 93.88 |
| SeTAR w $W_p$ | 3.20 | 98.93 | 20.73 | 94.77 | 23.91 | 93.53 | 22.06 | 95.89 | 30.65 | 94.38 | 39.50 | 91.41 | 23.34 | 94.82 |
| SeTAR w/o $W_p$ | 3.66 | 98.96 | 21.93 | 94.81 | 25.04 | 93.62 | 20.35 | 96.36 | 31.47 | 94.31 | 40.70 | 91.19 | 23.86 | 94.87 |

Table 17: **Detail results for SeTAR with different backbones.** [†] is cited from Jiang et al. (2024). [*] denotes the result of our re-run.

| Method | iNaturalist | | SUN | | Places | | Texture | | ImageNet22K | | COCO | | Average | |
|---|---|---|---|---|---|---|---|---|---|---|---|---|---|---|
| | FPR↓ | AUC↑ | FPR↓ | AUC↑ | FPR↓ | AUC↑ | FPR↓ | AUC↑ | FPR↓ | AUC↑ | FPR↓ | AUC↑ | FPR↓ | AUC↑ |
| **ImageNet1K** | | | | | | | | | | | | | | |
| **CLIP-base** | | | | | | | | | | | | | | |
| Vanilla MCM[*] | 32.07 | 94.43 | 38.65 | 92.37 | 43.73 | 90.03 | 57.89 | 86.13 | - | - | - | - | 43.09 | 90.74 |
| SeTAR+MCM | 26.92 | 94.67 | 35.57 | 92.79 | 42.64 | 90.16 | 55.83 | 86.58 | - | - | - | - | 40.24 | 91.05 |
| Vanilla GL-MCM[*] | 15.34 | 96.62 | 30.65 | 93.01 | 37.76 | 90.07 | 57.41 | 83.73 | - | - | - | - | 35.29 | 90.86 |
| SeTAR+GL-MCM | 13.36 | 96.92 | 28.17 | 93.36 | 36.80 | 90.40 | 54.17 | 84.59 | - | - | - | - | 33.12 | 91.32 |
| Vanilla NegLabel[†] | 1.91 | 99.49 | 20.53 | 95.49 | 35.59 | 91.64 | 43.56 | 90.22 | - | - | - | - | 25.40 | 94.21 |
| SeTAR+NegLabel | 0.15 | 99.54 | 19.06 | 95.84 | 30.63 | 92.22 | 42.54 | 90.30 | - | - | - | - | 23.09 | 94.48 |
| **CLIP-large** | | | | | | | | | | | | | | |
| Vanilla MCM[*] | 28.17 | 94.97 | 29.18 | 94.12 | 33.66 | 92.37 | 57.73 | 85.46 | - | - | - | - | 37.19 | 91.73 |
| SeTAR+MCM | 26.96 | 95.14 | 27.12 | 94.54 | 32.04 | 92.55 | 58.90 | 85.45 | - | - | - | - | 36.26 | 91.92 |
| Vanilla GL-MCM[*] | 29.58 | 94.43 | 32.54 | 93.35 | 37.18 | 91.43 | 63.28 | 80.71 | - | - | - | - | 40.65 | 89.98 |
| SeTAR+GL-MCM | 30.96 | 94.04 | 28.72 | 94.08 | 34.58 | 91.89 | 63.90 | 80.89 | - | - | - | - | 39.54 | 90.22 |
| **SwinTransformerV2-base** | | | | | | | | | | | | | | |
| Vanilla MSP[*] | 44.78 | 89.89 | 63.12 | 82.81 | 67.07 | 81.45 | 62.04 | 82.33 | - | - | - | - | 59.25 | 84.12 |
| SeTAR+MSP | 41.44 | 91.08 | 60.05 | 85.04 | 64.31 | 83.70 | 58.39 | 83.26 | - | - | - | - | 56.05 | 85.77 |
| Vanilla Energy[*] | 57.52 | 81.60 | 71.98 | 72.93 | 76.90 | 68.90 | 53.65 | 80.96 | - | - | - | - | 65.01 | 76.10 |
| SeTAR+Energy | 41.71 | 89.42 | 56.53 | 83.29 | 62.84 | 80.20 | 45.37 | 84.76 | - | - | - | - | 51.61 | 84.42 |
| **Pascal-VOC** | | | | | | | | | | | | | | |
| **CLIP-base** | | | | | | | | | | | | | | |
| Vanilla MCM[*] | 7.24 | 98.23 | 27.91 | 94.56 | 32.40 | 92.45 | 51.61 | 91.89 | 50.60 | 91.42 | 53.70 | 89.30 | 37.24 | 92.98 |
| SeTAR+MCM | 4.59 | 98.71 | 24.91 | 95.15 | 28.46 | 93.21 | 40.44 | 93.58 | 48.25 | 92.08 | 48.10 | 89.70 | 32.46 | 93.74 |
| Vanilla GL-MCM[*] | 4.33 | 98.81 | 22.94 | 94.63 | 26.20 | 93.11 | 41.61 | 92.88 | 37.88 | 93.17 | 43.70 | 90.71 | 29.44 | 93.88 |
| SeTAR+GL-MCM | 3.66 | 98.96 | 21.93 | 94.81 | 25.04 | 93.62 | 20.35 | 96.36 | 31.47 | 94.31 | 40.70 | 91.19 | 23.86 | 94.87 |
| **CLIP-large** | | | | | | | | | | | | | | |
| Vanilla MCM[*] | 42.90 | 94.69 | 44.27 | 93.28 | 41.48 | 91.57 | 61.33 | 89.95 | 63.37 | 91.20 | 59.90 | 89.40 | 52.21 | 91.68 |
| SeTAR+MCM | 26.05 | 96.23 | 35.97 | 94.20 | 33.10 | 92.45 | 50.32 | 91.91 | 57.69 | 92.02 | 52.30 | 90.67 | 42.57 | 92.91 |
| Vanilla GL-MCM[*] | 23.29 | 96.17 | 40.76 | 93.49 | 41.23 | 91.69 | 54.98 | 89.60 | 53.19 | 92.67 | 50.30 | 91.09 | 43.96 | 92.45 |
| SeTAR+GL-MCM | 9.62 | 97.51 | 27.75 | 94.73 | 28.85 | 92.99 | 41.77 | 92.40 | 39.42 | 93.98 | 39.30 | 92.38 | 31.12 | 94.00 |

Table 18: **Detail results for different search algorithms.** Here LES stands for layer-exhaustive greedy search, MIS stands for modality-interleave greedy search, and SeTAR-S stands for the search algorithm of SeTAR, which searches vision and text layers sequentially. We use CLIP-B/16 as a backbone.

| Method | iNaturalist | | SUN | | Places | | Texture | | ImageNet22K | | COCO | | Average | |
|---|---|---|---|---|---|---|---|---|---|---|---|---|---|---|
| | FPR↓ | AUC↑ | FPR↓ | AUC↑ | FPR↓ | AUC↑ | FPR↓ | AUC↑ | FPR↓ | AUC↑ | FPR↓ | AUC↑ | FPR↓ | AUC↑ |
| **ImageNet1K** | | | | | | | | | | | | | | |
| **MCM Score** | | | | | | | | | | | | | | |
| LES | 30.25 | 94.26 | 36.42 | 92.79 | 42.97 | 90.15 | 58.33 | 85.89 | - | - | - | - | 41.99 | 90.78 |
| MIS | 28.63 | 94.46 | 35.41 | 92.80 | 42.37 | 90.17 | 55.78 | 86.59 | - | - | - | - | 40.55 | 91.00 |
| SeTAR-S | 26.92 | 94.67 | 35.57 | 92.79 | 42.64 | 90.16 | 55.83 | 86.58 | - | - | - | - | 40.24 | 91.05 |
| **GL-MCM Score** | | | | | | | | | | | | | | |
| LES | 14.43 | 96.61 | 27.81 | 93.49 | 36.16 | 90.51 | 57.20 | 83.72 | - | - | - | - | 33.90 | 91.08 |
| MIS | 14.14 | 96.76 | 28.28 | 93.39 | 36.86 | 90.39 | 54.15 | 84.64 | - | - | - | - | 33.36 | 91.29 |
| SeTAR-S | 13.36 | 96.92 | 28.17 | 93.36 | 36.80 | 90.40 | 54.17 | 84.59 | - | - | - | - | 33.12 | 91.32 |
| **Pascal-VOC** | | | | | | | | | | | | | | |
| **MCM Score** | | | | | | | | | | | | | | |
| LES | 5.20 | 98.73 | 26.88 | 95.03 | 30.78 | 92.93 | 44.73 | 93.35 | 50.98 | 91.97 | 52.10 | 89.61 | 35.11 | 93.60 |
| MIS | 5.82 | 98.49 | 25.52 | 95.04 | 30.10 | 92.98 | 43.95 | 93.06 | 50.00 | 92.06 | 48.20 | 89.84 | 33.93 | 93.58 |
| SeTAR-S | 4.59 | 98.71 | 24.91 | 95.15 | 28.46 | 93.21 | 40.44 | 93.58 | 48.25 | 92.08 | 48.10 | 89.70 | 32.46 | 93.74 |
| **GL-MCM Score** | | | | | | | | | | | | | | |
| LES | 3.89 | 98.87 | 21.56 | 94.56 | 24.70 | 93.32 | 23.35 | 95.80 | 32.99 | 93.82 | 40.40 | 91.03 | 24.48 | 94.57 |
| MIS | 3.53 | 98.95 | 20.87 | 94.77 | 24.30 | 93.47 | 19.91 | 96.24 | 29.59 | 94.40 | 39.00 | 91.21 | 22.87 | 94.84 |
| SeTAR-S | 3.66 | 98.96 | 21.93 | 94.81 | 25.04 | 93.62 | 20.35 | 96.36 | 31.47 | 94.31 | 40.70 | 91.19 | 23.86 | 94.87 |

Table 19: **Detail results of ablation study on** $\lambda$**.** We use CLIP-B/16 as a backbone.

| $\lambda$ | iNaturalist | | SUN | | Places | | Texture | | ImageNet22K | | COCO | | **Average** | |
|---|---|---|---|---|---|---|---|---|---|---|---|---|---|---|
| | FPR↓ | AUC↑ | FPR↓ | AUC↑ | FPR↓ | AUC↑ | FPR↓ | AUC↑ | FPR↓ | AUC↑ | FPR↓ | AUC↑ | FPR↓ | AUC↑ |
| **ImageNet1K** | | | | | | | | | | | | | | |
| **MCM Score** | | | | | | | | | | | | | | |
| 0.01 | 28.31 | 94.60 | 36.83 | 92.74 | 43.01 | 90.10 | 55.85 | 86.58 | - | - | - | - | 41.00 | 91.00 |
| 0.05 | 27.41 | 94.75 | 35.91 | 92.70 | 42.75 | 90.15 | 55.57 | 86.49 | - | - | - | - | 40.41 | 91.02 |
| 0.10 | 26.92 | 94.67 | 35.57 | 92.79 | 42.64 | 90.16 | 55.83 | 86.58 | - | - | - | - | 40.24 | 91.05 |
| 0.15 | 34.29 | 93.66 | 35.88 | 92.85 | 42.34 | 90.24 | 58.09 | 86.01 | - | - | - | - | 42.65 | 90.69 |
| 0.20 | 34.89 | 93.62 | 35.59 | 92.88 | 41.95 | 90.28 | 58.19 | 86.11 | - | - | - | - | 42.66 | 90.72 |
| 0.25 | 35.88 | 93.42 | 35.48 | 92.76 | 42.24 | 90.18 | 58.39 | 85.84 | - | - | - | - | 43.00 | 90.55 |
| 0.30 | 37.72 | 93.26 | 36.27 | 92.64 | 42.35 | 90.10 | 58.46 | 86.03 | - | - | - | - | 43.70 | 90.50 |
| **GL-MCM Score** | | | | | | | | | | | | | | |
| 0.01 | 13.98 | 96.76 | 29.20 | 93.17 | 37.56 | 90.09 | 54.10 | 84.47 | - | - | - | - | 33.71 | 91.12 |
| 0.05 | 13.90 | 96.79 | 28.84 | 93.24 | 37.25 | 90.32 | 54.20 | 84.33 | - | - | - | - | 33.55 | 91.17 |
| 0.10 | 13.36 | 96.92 | 28.17 | 93.36 | 36.80 | 90.40 | 54.17 | 84.59 | - | - | - | - | 33.12 | 91.32 |
| 0.15 | 16.85 | 96.12 | 26.99 | 93.72 | 35.14 | 90.74 | 56.79 | 83.90 | - | - | - | - | 33.94 | 91.12 |
| 0.20 | 17.21 | 96.10 | 27.12 | 93.70 | 35.31 | 90.72 | 57.22 | 83.89 | - | - | - | - | 34.21 | 91.10 |
| 0.25 | 18.30 | 95.87 | 27.55 | 93.64 | 36.06 | 90.58 | 58.28 | 83.70 | - | - | - | - | 35.05 | 90.95 |
| 0.30 | 17.95 | 95.98 | 27.91 | 93.63 | 36.14 | 90.53 | 57.59 | 84.03 | - | - | - | - | 34.90 | 91.04 |
| **Pascal-VOC** | | | | | | | | | | | | | | |
| **MCM Score** | | | | | | | | | | | | | | |
| 0.01 | 5.58 | 98.43 | 25.14 | 94.94 | 29.13 | 93.01 | 40.41 | 93.35 | 47.85 | 92.12 | 49.60 | 89.37 | 32.95 | 93.54 |
| 0.05 | 4.59 | 98.71 | 24.91 | 95.15 | 28.46 | 93.21 | 40.44 | 93.58 | 48.25 | 92.08 | 48.10 | 89.70 | 32.46 | 93.74 |
| 0.10 | 5.44 | 98.50 | 24.97 | 95.06 | 29.60 | 93.01 | 42.55 | 93.26 | 48.69 | 92.28 | 47.80 | 89.82 | 33.18 | 93.65 |
| 0.15 | 5.97 | 98.53 | 26.50 | 95.07 | 30.88 | 93.05 | 46.22 | 92.94 | 50.99 | 92.07 | 49.80 | 89.80 | 35.06 | 93.58 |
| 0.20 | 6.11 | 98.53 | 26.18 | 95.08 | 30.53 | 93.06 | 45.43 | 93.06 | 50.68 | 92.16 | 49.40 | 89.82 | 34.72 | 93.62 |
| 0.25 | 6.41 | 98.43 | 26.19 | 94.99 | 31.24 | 92.89 | 47.36 | 92.72 | 50.41 | 92.13 | 50.20 | 89.74 | 35.30 | 93.48 |
| 0.30 | 6.81 | 98.34 | 26.98 | 94.80 | 32.13 | 92.65 | 48.67 | 92.52 | 50.53 | 92.14 | 51.10 | 89.77 | 36.04 | 93.37 |
| **GL-MCM Score** | | | | | | | | | | | | | | |
| 0.01 | 4.42 | 98.83 | 22.72 | 94.73 | 25.93 | 93.51 | 22.07 | 96.22 | 32.62 | 94.27 | 43.50 | 90.91 | 25.21 | 94.74 |
| 0.05 | 3.66 | 98.96 | 21.93 | 94.81 | 25.04 | 93.62 | 20.35 | 96.36 | 31.47 | 94.31 | 40.70 | 91.19 | 23.86 | 94.87 |
| 0.10 | 3.79 | 98.94 | 21.40 | 94.76 | 25.05 | 93.49 | 20.74 | 96.29 | 30.42 | 94.48 | 40.00 | 91.20 | 23.57 | 94.86 |
| 0.15 | 3.50 | 98.98 | 20.83 | 94.84 | 24.34 | 93.55 | 20.57 | 96.20 | 29.84 | 94.42 | 38.50 | 91.25 | 22.93 | 94.87 |
| 0.20 | 3.50 | 98.94 | 20.72 | 94.74 | 24.13 | 93.48 | 19.95 | 96.28 | 29.22 | 94.46 | 38.60 | 91.19 | 22.69 | 94.85 |
| 0.25 | 4.14 | 98.96 | 21.54 | 94.85 | 25.37 | 93.54 | 23.37 | 96.14 | 32.18 | 94.51 | 40.30 | 91.44 | 24.48 | 94.90 |
| 0.30 | 4.15 | 98.90 | 21.40 | 94.63 | 25.16 | 93.33 | 23.01 | 96.03 | 31.02 | 94.44 | 38.90 | 91.40 | 23.94 | 94.79 |

Table 20: **Detail results on different pruning strategies.** We use CLIP-B/16 as a backbone.

| Method | iNaturalist | | SUN | | Places | | Texture | | ImageNet22K | | COCO | | **Average** | |
|---|---|---|---|---|---|---|---|---|---|---|---|---|---|---|
| | FPR↓ | AUC↑ | FPR↓ | AUC↑ | FPR↓ | AUC↑ | FPR↓ | AUC↑ | FPR↓ | AUC↑ | FPR↓ | AUC↑ | FPR↓ | AUC↑ |
| **ImageNet1K** | | | | | | | | | | | | | | |
| **MCM Score** | | | | | | | | | | | | | | |
| Principle | 32.07 | 94.43 | 38.65 | 92.37 | 43.73 | 90.03 | 57.89 | 86.13 | - | - | - | - | 43.09 | 90.74 |
| Random | 32.07 | 94.43 | 38.65 | 92.37 | 43.73 | 90.03 | 57.89 | 86.13 | - | - | - | - | 43.09 | 90.74 |
| Minor | 26.92 | 94.67 | 35.57 | 92.79 | 42.64 | 90.16 | 55.83 | 86.58 | - | - | - | - | 40.24 | 91.05 |
| **GL-MCM Score** | | | | | | | | | | | | | | |
| Principle | 15.34 | 96.62 | 30.65 | 93.01 | 37.76 | 90.07 | 57.41 | 83.73 | - | - | - | - | 35.29 | 90.86 |
| Random | 32.07 | 94.43 | 38.65 | 92.37 | 43.73 | 90.03 | 57.89 | 86.13 | - | - | - | - | 43.09 | 90.74 |
| Minor | 13.36 | 96.92 | 28.17 | 93.36 | 36.80 | 90.40 | 54.17 | 84.59 | - | - | - | - | 33.12 | 91.32 |
| **Pascal-VOC** | | | | | | | | | | | | | | |
| **MCM Score** | | | | | | | | | | | | | | |
| Principle | 9.91 | 98.01 | 29.24 | 93.91 | 32.89 | 92.30 | 54.43 | 90.30 | 53.53 | 91.07 | 49.20 | 89.07 | 38.20 | 92.44 |
| Random | 7.24 | 98.20 | 27.45 | 94.60 | 32.52 | 92.43 | 43.30 | 93.25 | 49.89 | 91.02 | 52.97 | 89.06 | 35.57 | 93.09 |
| Minor | 4.59 | 98.71 | 24.91 | 95.15 | 28.46 | 93.21 | 40.44 | 93.58 | 48.25 | 92.08 | 48.10 | 89.70 | 32.46 | 93.74 |
| **GL-MCM Score** | | | | | | | | | | | | | | |
| Principle | 3.10 | 98.62 | 20.07 | 94.41 | 22.33 | 93.38 | 38.53 | 92.19 | 31.61 | 93.07 | 36.50 | 90.34 | 25.36 | 93.67 |
| Random | 3.47 | 98.99 | 20.04 | 95.46 | 24.07 | 93.95 | 31.76 | 94.86 | 35.71 | 93.67 | 42.17 | 91.04 | 26.20 | 94.66 |
| Minor | 3.66 | 98.96 | 21.93 | 94.81 | 25.04 | 93.62 | 20.35 | 96.36 | 31.47 | 94.31 | 40.70 | 91.19 | 23.86 | 94.87 |

Table 21: **Detail results of ablation study on top-K.** We use CLIP-B/16 as a backbone.

| K | iNaturalist | | SUN | | Places | | Texture | | ImageNet22K | | COCO | | **Average** | |
|---|---|---|---|---|---|---|---|---|---|---|---|---|---|---|
| | FPR↓ | AUC↑ | FPR↓ | AUC↑ | FPR↓ | AUC↑ | FPR↓ | AUC↑ | FPR↓ | AUC↑ | FPR↓ | AUC↑ | FPR↓ | AUC↑ |
| **ImageNet1K** | | | | | | | | | | | | | | |
| **MCM Score** | | | | | | | | | | | | | | |
| 0 | 26.50 | 94.70 | 36.22 | 92.66 | 43.04 | 90.10 | 55.82 | 86.46 | - | - | - | - | 40.39 | 90.98 |
| 100 | 26.92 | 94.67 | 35.57 | 92.79 | 42.64 | 90.16 | 55.83 | 86.58 | - | - | - | - | 40.24 | 91.05 |
| 200 | 26.92 | 94.67 | 35.57 | 92.79 | 42.64 | 90.16 | 55.83 | 86.58 | - | - | - | - | 40.24 | 91.05 |
| 300 | 26.92 | 94.67 | 35.57 | 92.79 | 42.64 | 90.16 | 55.83 | 86.58 | - | - | - | - | 40.24 | 91.05 |
| 400 | 26.92 | 94.67 | 35.57 | 92.79 | 42.64 | 90.16 | 55.83 | 86.58 | - | - | - | - | 40.24 | 91.05 |
| 500 | 26.92 | 94.67 | 35.57 | 92.79 | 42.64 | 90.16 | 55.83 | 86.58 | - | - | - | - | 40.24 | 91.05 |
| 600 | 26.50 | 94.70 | 36.22 | 92.66 | 43.04 | 90.10 | 55.82 | 86.46 | - | - | - | - | 40.39 | 90.98 |
| 700 | 26.92 | 94.67 | 35.57 | 92.79 | 42.64 | 90.16 | 55.83 | 86.58 | - | - | - | - | 40.24 | 91.05 |
| 800 | 26.92 | 94.67 | 35.57 | 92.79 | 42.64 | 90.16 | 55.83 | 86.58 | - | - | - | - | 40.24 | 91.05 |
| 900 | 29.38 | 94.38 | 36.02 | 92.75 | 42.47 | 90.24 | 55.20 | 86.77 | - | - | - | - | 40.77 | 91.03 |
| 1000 | 30.63 | 94.17 | 36.24 | 92.93 | 42.58 | 90.24 | 56.84 | 86.34 | - | - | - | - | 41.57 | 90.92 |
| **GL-MCM Score** | | | | | | | | | | | | | | |
| 0 | 14.02 | 96.80 | 28.32 | 93.40 | 36.91 | 90.52 | 54.68 | 84.32 | - | - | - | - | 33.48 | 91.26 |
| 100 | 13.36 | 96.92 | 28.17 | 93.36 | 36.80 | 90.40 | 54.17 | 84.59 | - | - | - | - | 33.12 | 91.32 |
| 200 | 13.36 | 96.92 | 28.17 | 93.36 | 36.80 | 90.40 | 54.17 | 84.59 | - | - | - | - | 33.12 | 91.32 |
| 300 | 13.36 | 96.92 | 28.17 | 93.36 | 36.80 | 90.40 | 54.17 | 84.59 | - | - | - | - | 33.12 | 91.32 |
| 400 | 13.36 | 96.92 | 28.17 | 93.36 | 36.80 | 90.40 | 54.17 | 84.59 | - | - | - | - | 33.12 | 91.32 |
| 500 | 13.36 | 96.92 | 28.17 | 93.36 | 36.80 | 90.40 | 54.17 | 84.59 | - | - | - | - | 33.12 | 91.32 |
| 600 | 14.02 | 96.80 | 28.32 | 93.40 | 36.91 | 90.52 | 54.68 | 84.32 | - | - | - | - | 33.48 | 91.26 |
| 700 | 13.36 | 96.92 | 28.17 | 93.36 | 36.80 | 90.40 | 54.17 | 84.59 | - | - | - | - | 33.12 | 91.32 |
| 800 | 13.36 | 96.92 | 28.17 | 93.36 | 36.80 | 90.40 | 54.17 | 84.59 | - | - | - | - | 33.12 | 91.32 |
| 900 | 14.71 | 96.63 | 28.64 | 93.31 | 36.56 | 90.41 | 54.04 | 84.78 | - | - | - | - | 33.49 | 91.28 |
| 1000 | 15.82 | 96.42 | 28.61 | 93.46 | 37.20 | 90.40 | 54.75 | 84.35 | - | - | - | - | 34.10 | 91.16 |
| **Pascal-VOC** | | | | | | | | | | | | | | |
| **MCM Score** | | | | | | | | | | | | | | |
| 0 | 5.58 | 98.43 | 25.14 | 94.94 | 29.13 | 93.01 | 40.41 | 93.35 | 47.85 | 92.12 | 49.60 | 89.37 | 32.95 | 93.54 |
| 2 | 4.59 | 98.71 | 24.91 | 95.15 | 28.46 | 93.21 | 40.44 | 93.58 | 48.25 | 92.08 | 48.10 | 89.70 | 32.46 | 93.74 |
| 4 | 4.59 | 98.71 | 24.91 | 95.15 | 28.46 | 93.21 | 40.44 | 93.58 | 48.25 | 92.08 | 48.10 | 89.70 | 32.46 | 93.74 |
| 6 | 5.58 | 98.43 | 25.14 | 94.94 | 29.13 | 93.01 | 40.41 | 93.35 | 47.85 | 92.12 | 49.60 | 89.37 | 32.95 | 93.54 |
| 8 | 5.27 | 98.45 | 24.26 | 94.98 | 28.31 | 93.06 | 39.61 | 93.31 | 46.99 | 92.11 | 48.10 | 89.38 | 32.09 | 93.55 |
| 10 | 5.58 | 98.43 | 25.14 | 94.94 | 29.13 | 93.01 | 40.41 | 93.35 | 47.85 | 92.12 | 49.60 | 89.37 | 32.95 | 93.54 |
| 12 | 4.59 | 98.71 | 24.91 | 95.15 | 28.46 | 93.21 | 40.44 | 93.58 | 48.25 | 92.08 | 48.10 | 89.70 | 32.46 | 93.74 |
| 14 | 5.58 | 98.43 | 25.14 | 94.94 | 29.13 | 93.01 | 40.41 | 93.35 | 47.85 | 92.12 | 49.60 | 89.37 | 32.95 | 93.54 |
| **GL-MCM Score** | | | | | | | | | | | | | | |
| 0 | 4.42 | 98.83 | 22.72 | 94.73 | 25.93 | 93.51 | 22.07 | 96.22 | 32.62 | 94.27 | 43.50 | 90.91 | 25.21 | 94.74 |
| 2 | 3.66 | 98.96 | 21.93 | 94.81 | 25.04 | 93.62 | 20.35 | 96.36 | 31.47 | 94.31 | 40.70 | 91.19 | 23.86 | 94.87 |
| 4 | 3.66 | 98.96 | 21.93 | 94.81 | 25.04 | 93.62 | 20.35 | 96.36 | 31.47 | 94.31 | 40.70 | 91.19 | 23.86 | 94.87 |
| 6 | 4.42 | 98.83 | 22.72 | 94.73 | 25.93 | 93.51 | 22.07 | 96.22 | 32.62 | 94.27 | 43.50 | 90.91 | 25.21 | 94.74 |
| 8 | 4.47 | 98.84 | 22.76 | 94.79 | 25.99 | 93.56 | 22.39 | 96.19 | 32.85 | 94.27 | 43.30 | 90.95 | 25.29 | 94.76 |
| 10 | 4.42 | 98.83 | 22.72 | 94.73 | 25.93 | 93.51 | 22.07 | 96.22 | 32.62 | 94.27 | 43.50 | 90.91 | 25.21 | 94.74 |
| 12 | 3.66 | 98.96 | 21.93 | 94.81 | 25.04 | 93.62 | 20.35 | 96.36 | 31.47 | 94.31 | 40.70 | 91.19 | 23.86 | 94.87 |
| 14 | 4.42 | 98.83 | 22.72 | 94.73 | 25.93 | 93.51 | 22.07 | 96.22 | 32.62 | 94.27 | 43.50 | 90.91 | 25.21 | 94.74 |

Table 22: **Detail results of ResNet50.** We use ImageNet1K as the ID dataset. [†] is cited from Djurisic et al. (2023).

| Method | iNaturalist | | SUN | | Places | | Texture | | **Average** | |
|---|---|---|---|---|---|---|---|---|---|---|
| | FPR↓ | AUC↑ | FPR↓ | AUC↑ | FPR↓ | AUC↑ | FPR↓ | AUC↑ | FPR↓ | AUC↑ |
| Softmax[†] | 54.99 | 87.74 | 70.83 | 80.86 | 73.99 | 79.76 | 68.00 | 79.61 | 66.95 | 81.99 |
| Energy[†] | 55.72 | 89.95 | 59.26 | 85.89 | 64.92 | 82.86 | 53.72 | 85.99 | 58.41 | 86.17 |
| ReAct[†] | 20.38 | 96.22 | 24.20 | 94.20 | 33.85 | 91.58 | 47.30 | 89.80 | 31.43 | 92.95 |
| DICE[†] | 25.63 | 94.49 | 35.15 | 90.83 | 46.49 | 87.48 | 31.72 | 90.30 | 34.75 | 90.77 |
| ASH-P[†] | 44.57 | 92.51 | 52.88 | 88.35 | 61.79 | 61.79 | 42.06 | 89.70 | 50.32 | 89.04 |
| ASH-B[†] | 14.21 | 97.32 | 22.08 | 95.10 | 33.45 | 92.31 | 21.17 | 95.50 | 22.73 | 95.06 |
| ASH-S[†] | 11.49 | 97.87 | 27.98 | 94.02 | 39.78 | 90.98 | 11.93 | 97.60 | 22.80 | 95.12 |
| SeTAR | 10.08 | 98.11 | 27.68 | 94.15 | 39.22 | 91.24 | 12.54 | 97.51 | 22.38 | 95.25 |

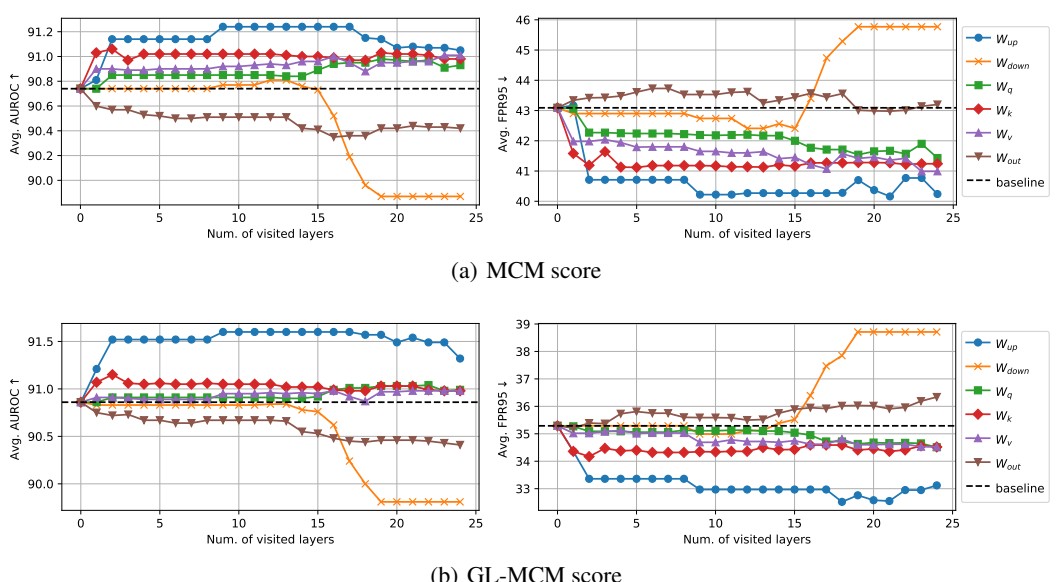

(a) MCM score

(b) GL-MCM score

Figure 2: **Average AUROC/FPR95 of different weight types on ImageNet1K benchmark.** We use CLIP-B/16 as a backbone.

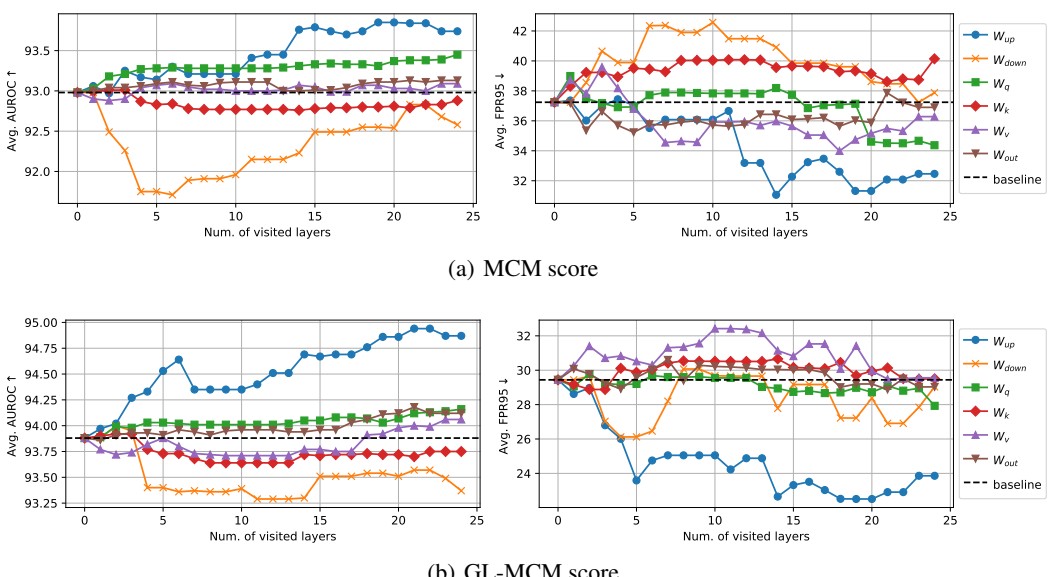

(a) MCM score

(b) GL-MCM score

Figure 3: **Average AUROC/FPR95 of different weight types on Pascal-VOC benchmark.** We use CLIP-B/16 as a backbone.

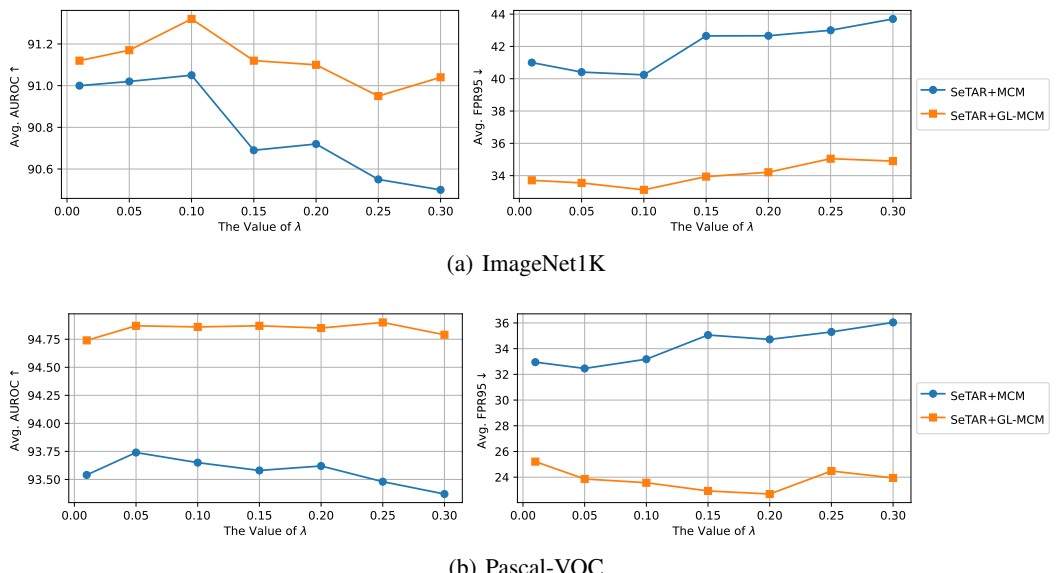

(a) ImageNet1K

(b) Pascal-VOC

Figure 4: **Ablation studies on $\lambda$ on different ID datasets.** We use CLIP-B/16 as a backbone.

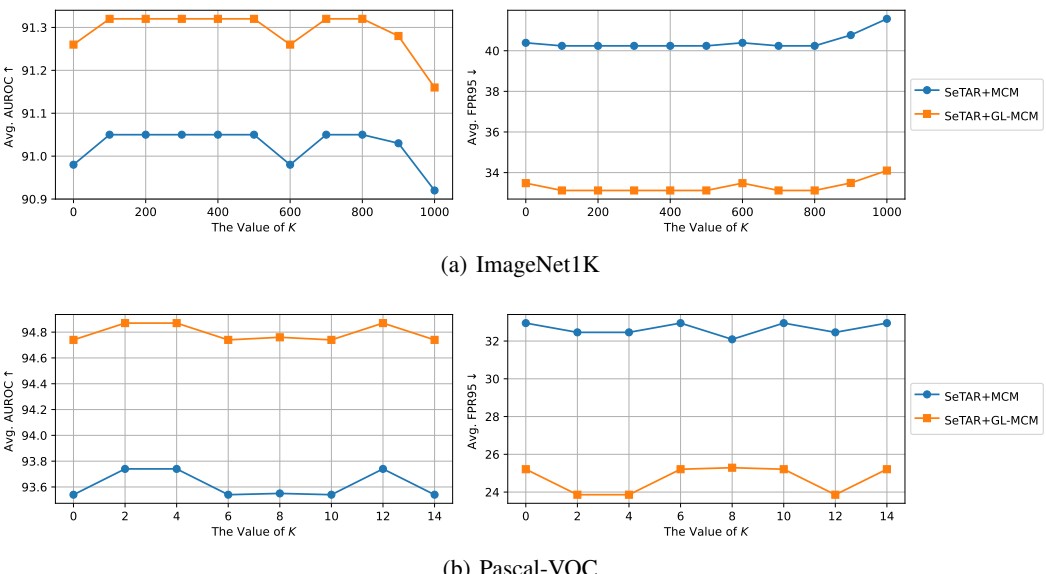

(a) ImageNet1K

(b) Pascal-VOC

Figure 5: **Ablation studies on top-K on different ID datasets.** We use CLIP-B/16 as a backbone.

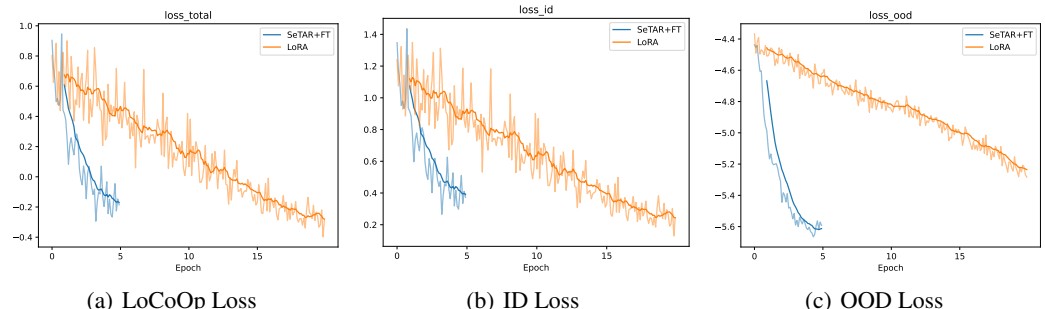

(a) LoCoOp Loss      (b) ID Loss      (c) OOD Loss

Figure 6: **Loss plots of SeTAR+FT v.s. LoRA on ImageNet1K.** We use CLIP-B/16 as a backbone. SeTAR+FT demonstrates faster convergence across all losses, especially in the OOD loss. For reference, with MCM score, SeTAR+FT achieves an average FPR of 38.77 at epoch 5. While LoRA achieves an average FPR of 42.88, 39.92 and 42.23 at epoch 1, 5 and 15, respectively.

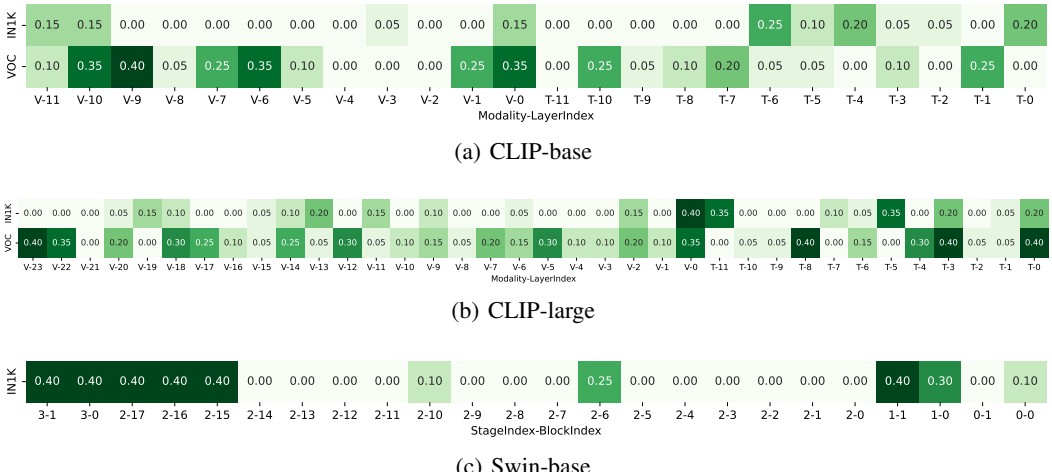

(a) CLIP-base

(b) CLIP-large

(c) Swin-base

Figure 7: **Visualization of SeTAR rank reduction ratio distribution on different ID datasets with different backbones.** IN1K, VOC stand for ImageNet1K and Pascal-VOC. And V, T stand for visual modality and text modality of the CLIP model.

```
      tower_type weight_type  layer_num  best_ratio  total_loss*   id_loss   ood_loss    val_acc   ood_patch_percent
step
0         visual        W_up         11        0.15     0.647777  1.093326  -4.455494  71.399998           38.906631
1         visual        W_up         10        0.15     0.644654  1.083629  -4.389751  71.799998           39.293876
2         visual        W_up          9        0.00     0.644654  1.083629  -4.389751  71.799998           39.293876
3         visual        W_up          8        0.00     0.644654  1.083629  -4.389751  71.799998           39.293876
4         visual        W_up          7        0.00     0.644654  1.083629  -4.389751  71.799998           39.293876
5         visual        W_up          6        0.00     0.644654  1.083629  -4.389751  71.799998           39.293876
6         visual        W_up          5        0.00     0.644654  1.083629  -4.389751  71.799998           39.293876
7         visual        W_up          4        0.00     0.644654  1.083629  -4.389751  71.799998           39.293876
8         visual        W_up          3        0.05     0.640844  1.079729  -4.388844  71.999998           39.209695
9         visual        W_up          2        0.00     0.640844  1.079729  -4.388844  71.999998           39.209695
10        visual        W_up          1        0.00     0.640844  1.079729  -4.388844  71.999998           39.209695
11        visual        W_up          0        0.15     0.640132  1.079109  -4.389775  72.199998           39.156123
12          text        W_up         11        0.00     0.640132  1.079109  -4.389775  72.199998           39.156123
13          text        W_up         10        0.00     0.640132  1.079109  -4.389775  72.199998           39.156123
14          text        W_up          9        0.00     0.640132  1.079109  -4.389775  72.199998           39.156123
15          text        W_up          8        0.00     0.640132  1.079109  -4.389775  72.199998           39.156123
16          text        W_up          7        0.00     0.640132  1.079109  -4.389775  72.199998           39.156123
17          text        W_up          6        0.25     0.630751  1.075123  -4.443716  71.600001           38.808673
18          text        W_up          5        0.10     0.630514  1.078703  -4.481889  71.599997           38.246428
19          text        W_up          4        0.20     0.622065  1.075958  -4.538932  72.000001           38.452552
20          text        W_up          3        0.05     0.620440  1.079326  -4.588857  71.999997           38.649488
21          text        W_up          2        0.05     0.618521  1.076858  -4.583368  71.600001           38.444899
22          text        W_up          1        0.00     0.618521  1.076858  -4.583368  71.600001           38.444899
23          text        W_up          0        0.20     0.615174  1.069851  -4.546776  72.499997           38.642345
```

Listing 1: **Example procedure of SeTAR on ImageNet1K with CLIP-base.** We search the visual and text tower from top to bottom. At each step, we select the best ratio that minimizes the loss.

