# OpenReview forum: "SeTAR: Out-of-Distribution Detection with Selective Low-Rank Approximation"
_NeurIPS.cc/2024/Conference — NeurIPS 2024 poster_

### Official Review · Reviewer_hoZJ · 2024-06-30

**Soundness:** 2
**Presentation:** 3
**Contribution:** 2
**Rating:** 6
**Confidence:** 4

**Summary:**

This work proposes SeTAR for CLIP-based OOD detection. SeTAR is based on low-rank approximation. It determines the optimal rank for each weight block with a greedy hyperparameter search using validation samples. While SeTAR itself is training-free, the paper also proposes SeTAR+FT which incorporates LoRA as a training extension. Experiments demonstrate that SeTAR outperforms existing methods on CLIP-specific benchmarks.

**Strengths:**

The paper is written in good quality, and the proposed method sounds reasonable.

**Weaknesses:**

1. My main concern is that the performance improvement of SeTAR (SeTAR+FT) seems limited compared with MCM and GL-MCM (LoCoOp). In Table 1, for example, all improvements of SeTAR in terms of AUROC are within 1%. The only case where SeTAR leads to noticeable/significant improvements is when the model is the image-only Swin-T (Table 2). Would the authors comment on why this is the case?

2. I'm confused by some of the numbers discussed in the main text. In line 220-222, it says `For example, using Pascal VOC as ID, SeTAR yields an average reduction of 12.84% FPR95 on MCM and 18.95% FPR95 on GL-MCM`. How is the "12.84" and "18.95" computed? According to Table 1, on Pascal VOC, the average FPR95 of MCM and SeTAR is 38.88 and 32.46, respectively. The average FPR95 of GL-MCM and SeTAR is 31.12 and 23.86, respectively. A similar confusion of mine is regarding line 238-240 `when scaled up to CLIP-large, SeTAR+FT outperforms LoCoOp and LoRA by 17.92% and 12.45% FPR95 on the same benchmark`, which doesn't seem to match with the numbers reported in Table 2. Am I missing anything?

3. All considered OOD datasets are far-OOD w.r.t. the ID set. Near-OOD detection has long been recognized as a more challenging and realistic problem [1, 2]. I would be very interested to see how well SeTAR performs on ImageNet v.s. NINCO or SSB [2].

I'm willing to adjust my score if my comments are addressed or clarified.

[1] Detecting semantic anomalies

[2] OpenOOD v1.5: Enhanced Benchmark for Out-of-Distribution Detection

**Questions:**

See weaknesses

**Limitations:**

Yes

---

> ### Author Rebuttal · Authors · 2024-08-07
>
> # 1. Performance Gains
>
> 1. **Limited Room for AUC Improvement:**
>    - The baseline AUC scores are above 90, leaving limited room for significant improvement. Despite this, our method still achieves AUC improvements, demonstrating its effectiveness even in a high-performance context.
>
> 2. **Significant FPR Improvement:**
>    - For the False Positive Rate (FPR), our method shows notable improvements. These reductions in FPR are significant and highlight the practical benefits of our approach in reducing false positives, enhancing the reliability of OOD detection.
>
>  3. **Swin-Base Performance:**
>     - The performance boost on Swin-Base is more pronounced because Swin is trained directly on ImageNet, lacking a text-encoder and thus requiring training solely on IN1K. This can lead to overfitting on ID data and poor recognition of OOD images, providing more room for improvement. Our method helps alleviate this issue, improving Swin's generalization to OOD samples.
>
>     - In contrast, CLIP models are pretrained on large image-text datasets, which provides robust representation capabilities for both ID and OOD images. Consequently, CLIP’s baseline OOD performance is already strong, leaving less room for further improvement compared to Swin-Base. Therefore, the pronounced performance boost seen with Swin-Base is due to its initial lower performance and higher potential for enhancement through our method.
>
> # 2. Numerical Clarifications
>
> 1. **Clarification on Pascal VOC FPR95 Reductions:**
>    - Mean MCM FPR95 reduction: (37.24 - 32.46) / 37.24 = 12.84%
>    - Mean GL-MCM FPR95 reduction: (29.44 - 23.86) / 29.44 = 18.95%
>
> 2. **Clarification on CLIP-large FPR95 Improvements:**
>    - Mean LoCoOp FPR95 improvement: (40.74 + 46.74 - 34.75 - 37.05) / (40.74 + 46.74) = 17.92%
>    - Mean LoRA FPR95 improvement: (38.62 + 43.39 - 34.75 - 37.05) / (38.62 + 43.39) = 12.45%
>
> These clarifications demonstrate the calculations behind the reported improvements, providing transparency and accuracy in our results.
>
> # 3. Near-OOD Results
>
> - We appreciate the suggestion and have added results for the CLIP-base backbone, with ImageNet1K as the ID dataset and SSB-Hard as the OOD dataset. SeTAR and SeTAR+FT show superior performance compared to the baselines.
>
>     | CLIP-base    | Category          |   MCM Score FPR↓ |   MCM Score AUC↑ |   GL-MCM Score FPR↓ |   GL-MCM Score AUC↑ |
>     |:-------------|:------------------|-----------------:|-----------------:|--------------------:|--------------------:|
>     | Vanilla      | Training-Free     |            89.28 |            63.88 |              85.62  |              67.63  |
>     | SeTAR        | Training-Free     |          **88.29** |          **64.20** |            **84.03** |            **68.29** |
>     | LoCoOp       | Finetuning (3run) |            89.72 |            63.45 |              86.79  |              65.93  |
>     | LoRA         | Finetuning (3run) |            88.52 |            65.38 |            **84.39** |              68.85  |
>     | SeTAR+FT     | Finetuning (3run) |          **87.16** |          **68.13** |              84.72  |            **70.42** |
>
> These results highlight SeTAR's and SeTAR+FT's robust performance in challenging near-OOD scenarios.

---

> ### Comment · Reviewer_hoZJ · 2024-08-08
> **Thanks for the rebuttal**
>
> My concerns are addressed well and I'd like to raise my score to 6. For point 2, please make it clear in the manuscript that the performance improvement is relative (by default I think people assume/expect absolute improvement being discussed). For point 3, I strongly recommend including the near-OOD results in the main text so that later works can continue to work the meaningful & challenging near-OOD detection problem (rather than keep working only on far-OOD by following previous works).

---

> > ### Author Response · Authors · 2024-08-08
> > **Thanks for the Feedback and Revised Score**
> >
> > Thank you very much for your constructive feedback and for raising the score. We are grateful for your detailed review and are glad that our responses addressed your concerns. We will ensure that the manuscript clearly indicates that the performance improvements are relative. Additionally, we will incorporate the near-OOD results into the main text as suggested. Your insights have been invaluable in enhancing our work, and we sincerely appreciate your support and encouragement.

---

### Official Review · Reviewer_6fvA · 2024-07-02

**Soundness:** 3
**Presentation:** 2
**Contribution:** 2
**Rating:** 5
**Confidence:** 4

**Summary:**

The paper introduces SeTAR, a novel training-free out-of-distribution (OOD) detection method that leverages selective low-rank approximation of weight matrices in vision-language and vision-only models. SeTAR enhances OOD detection by post-hoc modifying the model’s weight matrices using a greedy search algorithm. The paper also extends this method to SeTAR+FT, a fine-tuning approach to further optimize OOD detection performance.

**Strengths:**

1.	The paper proposes a unique training-free method for enhancing OOD detection, which is novel.
2.	The paper includes thorough ablation studies and sensitivity analyses, which help in understanding the robustness and generalizability of the proposed approach.
3.	The authors provide extensive empirical evaluations on multiple benchmarks, showing the effectiveness of SeTAR and SeTAR+FT.

**Weaknesses:**

-	The performance gains compared to baselines are relatively small. Given that these results are achieved using a greedy search strategy, the potential of SeTAR may be limited.
-	There is a lack of theoretical analysis explaining why SeTAR is effective. Understanding the underlying principles is crucial for advancing the method and its applications.
-	The experiments indicate that the optimal hyperparameters (λ and K) vary significantly across different backbones. In practical OOD detection scenarios, it is challenging to obtain OOD samples for hyperparameter tuning. Therefore, SeTAR needs to be strengthened in this aspect to ensure robust performance without extensive hyperparameter tuning.
P.S. The layout of Figure 1 is somewhat cluttered.

**Questions:**

Please refer to weaknesses.

**Limitations:**

Yes

---

> ### Author Rebuttal · Authors · 2024-08-07
>
> # 1. Performance Gains
>
> 1. **Limited Room for AUC Improvement:**
>    - The baseline AUC scores are above 90, leaving limited room for significant improvement. Despite this, our method still achieves AUC improvements, demonstrating its effectiveness even in a high-performance context.
>
> 2. **Significant FPR Improvement:**
>    - For the False Positive Rate (FPR), our method shows notable improvements. These reductions in FPR highlight the practical benefits of our approach in reducing false positives, enhancing the reliability of OOD detection.
>
> 3. **Pronounced Performance on Swin-Base:**
>    - As shown in Table 2 and Table 7, the performance boost on Swin-Base is significant. For example, using SeTAR with the Energy score on ImageNet1K, the reductions in FPR and improvements in AUC are over 20% and 9.8% compared to the baseline. SeTAR+FT further reduces FPR and improves AUC by more than 36% and 20% compared to the baseline.
>
> These points emphasize the substantial performance gains achieved by our method, particularly on Swin-Base, highlighting its effectiveness in various contexts.
>
> # 2. Theoretical Analysis of SeTAR's Effectiveness
>
> To address this, we draw on theoretical principles from recent work on SVD-based weight pruning, particularly from the study titled "[Enhancing In-Context Learning Performance with just SVD-Based Weight Pruning: A Theoretical Perspective](https://arxiv.org/pdf/2406.03768)".
>
> SeTAR employs selective low-rank approximation through SVD-based pruning, which aligns with established theoretical frameworks that explain how such techniques can enhance model performance:
>
> 1. **Gradient Stability and Generalization:**
>    - The theoretical analysis of SVD-based weight pruning shows that pruning can stabilize the gradient updates in neural networks. By pruning weights, particularly in layers where noise might be higher, we reduce the sensitivity of the network to small perturbations, leading to more stable and robust performance. This stability is crucial for OOD detection tasks, as it allows the model to maintain its performance across varying inputs.
>
> 2. **Matrix Condition Number and Noise Reduction:**
>    - The effectiveness of weight pruning can also be understood through the concept of matrix condition numbers. High condition numbers indicate ill-conditioned problems prone to significant errors due to small perturbations. Pruning minor singular values reduces the condition number, stabilizing the model and enhancing its robustness to noise. This is crucial for tasks like OOD detection, where stability and robustness are key.
>
> 3. **Retention of Principal Components:**
>    - By retaining the principal components (i.e., the components corresponding to the largest singular values), SeTAR ensures that the most critical information is preserved while reducing noise. This principle, rooted in the Eckart-Young-Mirsky theorem, provides an optimal low-rank approximation that maintains essential features necessary for effective OOD detection.
>
>
> # 3. Hyperparameter Tuning
>
> 1. **Robustness of Top-K Hyperparameter:**
>    - The optimal Top-K parameter is related to the number of ID categories and cannot be directly transferred between different ID datasets. However, as shown in Figure 6, this parameter is quite robust. We generally recommend setting it to 30% of the total number of categories. For example, ImageNet1K (300/1000) and Pascal-VOC (4/14). For the Swin-base model, setting Top-K to 300 also yields good performance.
>
>     | Backbone | Score | Vanilla Method FPR↓ | Vanilla Method AUC↑ | SeTAR (TopK700) FPR↓ | SeTAR (TopK700) AUC↑ | SeTAR (TopK300) FPR↓ | SeTAR (TopK300) AUC↑ |
>     |:-----|:--------|-------:|-------:|--------:|--------:|--------:|--------:|
>     | Swin-base| MSP | 59.25| 84.12| **56.05** | **85.77** | 56.82 | 85.68 |
>     | Swin-base| Energy| 65.01| 76.10| **51.61** | **84.42** | 52.56 | 84.51 |
>
> 2. **Transferability of λ Hyperparameter:**
>    - The λ parameter shows transferability across different datasets for the same backbone. As shown in Table 9, the λ for CLIP-base ranges between 0.05 and 0.1, while for CLIP-large it ranges between 0.3 and 0.5.
>
> 3. **Hyperparameter Transferability Across Datasets:**
>    - The optimal hyperparameters exhibit transferability across different datasets. For instance, the optimal hyperparameters for CLIP-Base on ImageNet1K and Pascal-VOC can be interchanged with minimal performance degradation, and both outperform the vanilla method.
>
>     | ImageNet1K | Score  | Vanilla Method FPR↓ | Vanilla Method AUC↑ | SeTAR Optimal hyperpara FPR↓ | SeTAR Optimal hyperpara AUC↑ | SeTAR with VOC optimal hyperpara FPR↓ | SeTAR with VOC optimal hyperpara AUC↑ |
>     |:--|:-------|---:|---:|---:|---:|---:|---:|
>     | CLIP-base  | MCM| 43.09| 90.74| **40.24**| **91.05**| 40.41 | 91.02 |
>     || GL-MCM | 35.29| 90.86| **33.12**| **91.32**| 33.55 | 91.17 |
>     | CLIP-large | MCM| 37.19| 91.73| **36.26**| **91.92**| 36.73 | 91.81 |
>     || GL-MCM | 40.65| 89.98| **39.54**| **90.22**| 39.18 | 90.10 |
>
>     | Pascal-VOC | Score| Vanilla Method FPR↓ | Vanilla Method AUC↑ | SeTAR Optimal hyperpara FPR↓ | SeTAR Optimal hyperpara AUC↑ | SeTAR with ImageNet1K optimal hyp FPR↓ | SeTAR with ImageNet1K optimal hyp AUC↑ |
>     |:-----------|:-------|------:|------:|--:|--:|------------:|------------:|
>     | CLIP-base| MCM| 37.24| 92.98| **32.46** | **93.74** | 33.18|93.65|
>     || GL-MCM | 29.44| 93.88| **23.86** | **94.87**| 23.57| 94.86|
>     | CLIP-large | MCM| 52.21| 91.68| **42.57** | **92.91** | 44.39| 92.34|
>     || GL-MCM | 43.96 | 92.45 | **31.12** | **94.00**| 33.74 | 93.76|
>
> 4. **Figure 1 Layout:**
>    - We appreciate the feedback on Figure 1 and have adjusted its layout for better clarity.

---

> > ### Comment · Reviewer_6fvA · 2024-08-10
> >
> > Thank you for the detailed reply.
> >
> > I find the explanation about the "limited room for AUC improvement" somewhat inadequate. Additionally, I noticed that your primary comparisons are with MCM and its variant GL-MCM. I suggest comparing SeTAR with more recent baselines, such as “CLIPN for Zero-Shot OOD Detection: Teaching CLIP to Say No” (ICCV2023) and “Negative Label Guided OOD Detection with Pretrained Vision-Language Models” (ICLR2024), to more convincingly demonstrate its performance advantages.
> >
> > Your responses to the theoretical analysis and hyperparameter tuning concerns have addressed some of my doubts. Thank you for the additional clarifications.

---

> ### Author Response · Authors · 2024-08-12
> **Comparisons with CLIPN and NegLabel**
>
> We apologize for the late reply and appreciate your patience.
>
> # 1. Comparisons with CLIPN
>
> - CLIPN [1] is pre-trained on the **CC-3M** dataset for **10 epochs**, involves training an additional NO-encoder for over 64 million parameters
> - On ViT-B/16, CLIPN-C achieves an FPR95 of 38.59 and an AUROC of 86.35, while CLIPN-A achieves an FPR95 of 31.10 and an AUROC of 93.10.
> - In contrast, our SeTAR method does not involve any training. When combined with fine-tuning (SeTAR+FT), it operates in a 1-shot setting, utilizing only **1,000 images** for training, with just **1.6% of the parameters** being trainable over **5 epochs**.
> - For comparison, SeTAR achieves an FPR95 of 33.12 and an AUROC of 91.32 without any training. With minimal fine-tuning (SeTAR+FT), it achieves an FPR95 of 32.19 and an AUROC of 92.31 with only 1,000 training samples.
>
> Given the significant differences in the amount of training data, the number of trainable parameters, and the computational resources required, we believe that a direct comparison between CLIPN and our method is not entirely fair. Despite the minimal computational resources involved, SeTAR and SeTAR+FT achieve performance levels close to those of CLIPN, demonstrating the efficiency of our approach.
>
> # 2. Comparisons with NegLabel
>
> - **Difference from NegLabel:** Our method differs from NegLabel [2] in its focus. SeTAR primarily aims to enhance the model's intrinsic performance through SVD pruning, without incorporating any additional knowledge or inputs. In contrast, NegLabel improves OOD detection by constructing large scale virtual negative labels from the data perspective.
> - **Compatibility with NegLabel:** Since SeTAR and NegLabel have different focuses, they are not mutually exclusive. As mentioned in our paper, SeTAR is highly compatible with various score functions and can also work alongside data augmentation methods like NegLabel. For instance, by simply merging negative labels into the label space for searching and testing, SeTAR can further surpass NegLabel's performance.
>
>     |ViT-B/16 |iNaturalist FPR↓|iNaturalist AUC↑|SUN FPR↓|SUN AUC↑|Places FPR↓|Places AUC↑|Texture FPR↓|Texture AUC↑|Average FPR↓|Average AUC↑|
>     |:---|-:|-:|-:|----:|-:|-:|-:|-:|-:|-:|
>     |CLIPN\*|23.94 |95.27|26.17|93.93|33.45|92.28|40.83|90.93|31.10|93.10|
>     |NegLabel\* |1.91 |99.49|20.53|95.49|35.59|91.64|43.56|90.22|25.40|94.21|
>     |SeTAR|0.15 |99.54|19.06|95.84|30.63|92.22|42.54|90.30|**23.09**|**94.48**|
>
>     > \* cited from [2]
>
> In summary, SeTAR demonstrates strong performance through its structural enhancements, particularly with SVD pruning, which significantly improves model robustness and OOD detection without requiring additional training. Furthermore, when combined with data augmentation methods like NegLabel, SeTAR's effectiveness is further amplified, showing even greater improvements in OOD detection metrics.
>
> ---
>
> **References**:
>
> - [1] CLIPN for Zero-Shot OOD Detection: Teaching CLIP to Say No
> - [2] Negative Label Guided OOD Detection with Pretrained Vision-Language Models

---

> > ### Author Response · Authors · 2024-08-13
> > **Anticipating Further Remarks**
> >
> > Dear Reviewer 6fvA
> >
> > As the discussion period is coming to a close, we would appreciate it if you could let us know whether our recent rebuttal has addressed some of your concerns or questions. We are more than happy to address any further issues you may have. Engaging in this discussion will greatly help us in refining and improving our paper.
> >
> > We look forward to your response or acknowledgment once you have read our message, as your support is very important to us.
> >
> > Best regards,
> >
> > The Authors

---

> > > ### Comment · Reviewer_6fvA · 2024-08-13
> > >
> > > Thank you for your response. I have read the other comments as well as all your rebuttals. The contribution of this paper is mainly focused on OOD detection tasks based on CLIP. However, SeTAR faces certain challenges when compared to CLIPN and NegLabel. I still have concerns about the relatively limited performance of SeTAR. I believe this paper is a borderline (4 or 5). I may raise the rating to 5 after thorough discussion in the next phase.

---

> > > > ### Author Response · Authors · 2024-08-14
> > > > **SeTAR’s Distinct Advantages and Compatibility**
> > > >
> > > > # SeTAR’s Distinct Advantages and Compatibility
> > > >
> > > > We are pleased to learn that our previous rebuttal successfully addressed many of your concerns, and we sincerely appreciate your thoughtful feedback throughout this process. However, we would like to take this opportunity to further clarify some key aspects regarding the performance of SeTAR, particularly in comparison to existing methods like CLIPN and NegLabel.
> > > >
> > > > # 1. **New Perspective and Orthogonality:**
> > > >
> > > > - Our method enhances OOD performance from a novel angle and is designed to be **highly compatible with a wide range of OOD methods**. SeTAR can be integrated with different score functions (MCM, GL-MCM, MSP, Energy), various backbones (CLIP, Swin, ResNet), and advanced OOD methods like CLIPN and NegLabel. This flexibility underscores that our approach is not in conflict with existing OOD methods but rather offers a complementary solution that can be layered with them. **By focusing on improving the intrinsic capability of models**, SeTAR is positioned to continuously enhance the effectiveness of other potential OOD methods in the future.
> > > >
> > > > - In contract, both CLIPN and NegLabel enhance OOD detection by leveraging negative text labels, which **inherently limits their applicability to vision-language models like CLIP**. This approach is not directly applicable to pure vision models such as Swin and ResNet, which do not incorporate a text encoder. In contrast, our method is not constrained by such limitations and can be effectively applied to both vision-language models and pure vision models, offering a broader applicability across different model architectures.
> > > >
> > > > # 2. **Lightweight and Efficient Approach:**
> > > >
> > > > - SeTAR and SeTAR+FT are designed as **lightweight and efficient methods**. SeTAR specifically addresses zero-shot scenarios, while SeTAR+FT combines our approach with existing parameter-efficient fine-tuning methods to enable efficient training. In contrast, other methods often rely on extensive training or large amounts of external data.
> > > > - For instance, CLIPN involves training an additional NO-encoder using the CC-3M dataset, which is **3000 times** larger in data volume, **25 times** greater in trainable parameters, and requires over **300 thousand times** training time, compared to SeTAR+FT. Similarly, NegLabel constructs negative labels information from **70,000 label candidates**.
> > > > - In contrast, SeTAR operates independently of external training data such as CC-3M or additional labels, and requires no further training, underscoring its efficiency and practicality.
> > > >
> > > > # 3. **Advancing the State of the Art:**
> > > >
> > > > - Even when applied to a high-performing method like NegLabel, SeTAR further enhances model performance. Specifically, we were able to **reduce the average FPR by an additional 9% relative to NegLabel’s already impressive 25.40 average FPR**, achieving a new state-of-the-art with a 23.09 average FPR. This demonstrates SeTAR’s potential to push the boundaries of current OOD detection performance to new levels.
> > > >
> > > > We hope these clarifications further illustrate the value and uniqueness of SeTAR. We are grateful for your consideration and are committed to refining our work based on your insights.

---

> > > > > ### Comment · Reviewer_6fvA · 2024-08-14
> > > > >
> > > > > Thank you for your clarification. Overall, I believe this paper is a borderline. I will raise the rating to 5 and be open to the AC's judgment.

---

> > > > ### Author Response · Authors · 2024-08-14
> > > > **Thank You for Your Consideration and Feedback**
> > > >
> > > > Thank you very much for your thoughtful consideration and for raising the rating. We appreciate your constructive feedback throughout the review process. Your insights have been invaluable in helping us refine our work, and we are grateful for your support.

---

### Official Review · Reviewer_SyYP · 2024-07-03

**Soundness:** 3
**Presentation:** 3
**Contribution:** 3
**Rating:** 5
**Confidence:** 3

**Summary:**

This paper proposes an algorithm for OOD detection along with CLIP models.
It observes that pruning based on SVD decomposition on CLIP models can improve the OOD detection performance.
A greedy search algorithm is developed for searching pruning rations of each weight in CLIP models.

Experiments on regular settings demonstrate obvious performance when compared with the Vanilla GL_MCM method.

**Strengths:**

(1) The paper is written clearly and easy to follow.
(2) The method is simple but effective.
(3) The ablation studies are sound.

**Weaknesses:**

(1) It seems that the proposed method is general enough to apply to regular models not just CLIP models.
     Will it also work on CNN-based ResNets and ViT models trained on datasets, like ImageNet and CIFAR?
      The experimental results focus on CLIP-based models in Table 1.
     More comparisons with previous methods using CNN-based models should be included.

(2) SVD decomposition pruning has strong connections with sparsification-based methods [1,2,3].
     Comparisons with this kind of method should be included.

[1] Dice: leveraging sparsification for out-of-distribution detection.
[2] Extremely simple activation shaping for out-of-distribution detection.
[3] ReAct: out-of-distribution detection with rectified activations.

(3) The proposed greedy search algorithm is efficient. However, it can not be guaranteed to be converged.
     What's the performance variance for 2 independent runs of the search algorithm?

**Questions:**

(1) The main results in the paper concentrate on CLIP models.
      Results on other CNN-based models and comparisons with previous methods should be included in the paper.

(2) The proposed greedy search algorithm can not be guaranteed to be converged.
      Thus, a discussion on the stability of the proposed algorithm is required.

**Limitations:**

A section of Impact Statements including limitations is included in the paper

---

> ### Author Rebuttal · Authors · 2024-08-07
>
> # 1. Applicability to CNN-Models
>
> 1. **Will it work on CNN-based ResNets?**
>    - No. Our method is not applicable to pure CNN models like traditional ResNet50. The loss function (Eq. 12) in our approach includes an OOD loss (Eq. 11), which relies on local features from the attention layer. Since pure CNN models lack self-attention layers, our method cannot be directly used or compared with CNN-based models.
>    - However, in CLIP-ResNet models, a self-attention layer is added to the last layer of the ResNet tower. Therefore, we can conduct experiments on CLIP-ResNet models.
>    - Specifically, we used the CLIP-ResNet50x4 model as the backbone with ImageNet1K as the ID dataset. By applying SVD pruning on the conv1 layer of each vision layer and  W_{up} on each text layer, we obtained the following results, demonstrating that our method is also applicable to CLIP-ResNet models (\* stands for our re-run).
>
>         | CLIP-ResNet50x4  | iNaturalist FPR↓ | iNaturalist AUC↑ | SUN FPR↓ | SUN AUC↑ | Places FPR↓ | Places AUC↑ | Texture FPR↓ | Texture AUC↑ | Average FPR↓ | Average AUC↑ |
>         |:----------------|-------------------:|-------------------:|-----------:|-----------:|--------------:|--------------:|---------------:|---------------:|---------------:|---------------:|
>         | Vanilla MCM\*    | 44.03 | 91.58 | 35.18 | 92.83 | 44.38 | 89.38 | 57.29 | 85.99 | 45.22 | 89.95 |
>         | SeTAR+MCM        | 41.29 | 92.12 | 35.44 | 92.76 | 43.01 | 89.85 | 54.82 | 86.89 | **43.64** | **90.40** |
>         | Vanilla GL-MCM\* | 32.17 | 93.09 | 46.64 | 89.27 | 51.85 | 85.86 | 44.47 | 86.49 | 43.78 | 88.68 |
>         | SeTAR+GL-MCM     | 30.15 | 93.73 | 45.01 | 89.58 | 49.82 | 86.68 | 42.32 | 87.30 | **41.83** | **89.32** |
>
> 2. **Will it work on ViT models trained on datasets like ImageNet and CIFAR?**
>    - Yes. The Swin-Transformer is a ViT model that does not include a text-encoder but only an image-encoder. As shown in Table2 and Table 7, we used the Swin-base model trained on ImageNet1K as the backbone and observed significant improvements over the original model.
>
> 3. **More Comparisons with Previous Methods Using CNN-based Models:**
>    - As mentioned, our method is not applicable to CNN-based models. More relevant comparisons are addressed in the response to question 2.
>
> These points clarify that while our method is not suitable for pure CNN models, it is effective for models incorporating attention mechanisms, such as CLIP-ResNet and ViT models like Swin-Transformer.
>
> # 2. Comparisons with Sparsification-Based CNN Methods
>
> We appreciate the reviewer's suggestion and have carefully reviewed the referenced papers. These methods are based on CNN-ResNet models and the ImageNet1K dataset. However, there is a significant difference between these models and the CLIP-ResNet:
>
> 1. **Training Differences:**
>    - CNN-ResNet models, due to the lack of a text-encoder, are fine-tuned directly on ImageNet1K.
>    - In contrast, CLIP-ResNet models are not trained directly on ImageNet1K. Therefore, directly comparing the results of CLIP-ResNet with CNN-ResNet models is not meaningful, since ID-domain training would largely boost the model performance.
>
> 2. **Lack of Suitable CLIP-ResNet Models:**
>    - We attempted to find CLIP-ResNet models that were fine-tuned on ImageNet1K for a fair comparison but were unable to locate such models.
>
> 3. **Truly Training-Free Nature of Our Method:**
>    - Our method is **literally training-free**, as it does not require any training on the ID dataset. In contrast, the backbones used in the sparsification-based methods require training on the ID dataset.
>
> These points highlight the challenges in making direct comparisons with sparsification-based methods and underscore the unique, training-free advantage of our approach.
>
> # 3. Convergence and Performance Variance Concerns
>
> The SVD algorithm used in our method is quite stable. We have compared the results using different random seeds (3, 4, and 5), and the SVD results are consistent across these different seeds. Therefore, SeTAR does not involve performance variance due to the deterministic nature of the SVD algorithm used in our approach.

---

> > ### Comment · Reviewer_SyYP · 2024-08-08
> > **Further questions**
> >
> > Hi, thanks for the responses from the authors. I still have some confusion on the paper.
> > (1) Is there any alternative loss function for Eq. 11?
> >       Although the proposed method is general, its capability is heavily limited by this loss.
> >
> > (2) Could the authors provide comparisons with sparsification-based methods using Swin or ViT backbones?

---

> ### Author Response · Authors · 2024-08-10
> **Reply to further questions**
>
> We sincerely appreciate the thoughtful feedback provided by the reviewer. Here are our responses:
>
> # 1. Alternative Loss Function for Eq. 11
>
> - Our loss function (Eq. 11) leverages local features because it requires pseudo-OOD features, which are inherently present in ViTs and can also be constructed in CNNs. In CNNs, alternative methods for OOD feature construction are available, such as the approach proposed in [NPOS](https://arxiv.org/pdf/2303.02966)[4], where boundary ID embeddings are selected based on non-parametric k-NN distances, and outliers are synthesized by sampling from a multivariate Gaussian distribution centered around these boundary embeddings. Additionally, [CLIP-OS](https://arxiv.org/pdf/2404.00323)[5] suggests using CLIP for outlier synthesis, which can similarly be adapted for constructing OOD features in CNNs.
>
> - However, due to the complexity of implementation and time constraints, we were unable to conduct the corresponding experiments. Nevertheless, our method is not limited to CLIP models; by utilizing different OOD feature construction methods, our approach can be readily adapted to CNN models as well.
>
> # 2. Comparisons on Swin Backbone
>
> - We conducted experiments using the Swin backbone with ReAct [3], DICE [1], and ASH [2] methods. Specifically:
>
>     1. **Codebase:** For ASH and ReAct, we implemented Swin-transformer based on the [official ASH repository](https://github.com/andrijazz/ash/blob/main/config/vit_config.yml), strictly following the original settings and applying sparsification before the final linear layer. Since ASH does not provide an implementation for DICE, we based our implementation on the [DICE official repository](https://github.com/deeplearning-wisc/dice/tree/master/models).
>
>     2. **Hyperparameter Search:** Following ASH, we experimented with various parameter combinations and reported the best results: DICE pruning thresholds included {10%, 15%, 70%}, ReAct clipping thresholds {1.0, 1.5, 1.33}, and ASH-S pruning thresholds included {60%, 65%, 90%, 95%}.
>
>     3. **Datasets:** We used ImageNet1K as the ID dataset.
>
> - The results are shown below. It is evident that compared to sparsification-based methods, SeTAR exhibits the best performance. ReAct shows improvements over the baseline in both scoring functions, while DICE shows improvement only with MSP; ASH-S performs poorly across the board.
>
>     | SwinV2-Base| Score| iNaturalist FPR↓ | iNaturalist AUC↑ | SUN FPR↓ | SUN AUC↑ | Places FPR↓ | Places AUC↑ | Texture FPR↓ | Texture AUC↑ | Average FPR↓ | Average AUC↑ |
>     |:-|:-|-:|-:|-:|-:|-:|-:|-:|-:|-:|-:|
>     | Vanilla* | MSP|44.78 |89.89 |63.12 |82.81 | 67.07 | 81.45 |62.04 |82.33 |59.25 |84.12 |
>     | React* | MSP|42.98 |90.39 |61.34 |83.89 | 65.11 | 82.64 |61.22 |81.37 |57.66 |84.57 |
>     | DICE*| MSP|43.02 |89.03 |62.22 |78.31 | 65.82 | 79.35 |57.75 |81.48 |57.20 |82.04 |
>     | ASH-S* | MSP|53.21 |78.72 |73.71 |66.56 | 79.75 | 60.75 |60.27 |75.52 |66.73 |70.39 |
>     | SeTAR (Ours) | MSP| **41.44** |**91.08** |**60.05** |**85.04** | **64.31** | **83.70** |**58.39** |**83.26** |**56.05** |**85.77** |
>     | Vanilla* | Energy |57.52 |81.60 |71.98 |72.93 | 76.90 | 68.90 |53.65 |80.96 |65.01 |76.10 |
>     | React* | Energy |41.78 |88.34 |60.98 |79.19 | 68.07 | 75.56 |53.72|80.39|56.14|80.87|
>     | DICE*| Energy |64.45 |74.91|83.04|62.67|95.18|47.05|94.65|33.45|84.33|58.18|
>     | ASH-S* | Energy |99.67 |19.26 |99.29 |22.91 | 99.51 | 21.49 |98.51 |33.45 |99.25 |24.28 |
>     | SeTAR (Ours) | Energy |**41.71** |**89.42** |**56.53** |**83.29** | **62.84** | **80.20** |**45.37** |**84.76** |**51.61** |**84.42** |
>
>     > \* denotes our rerun
>
> These results demonstrate that SeTAR significantly outperforms sparsification-based methods, highlighting the effectiveness of our approach.
>
> # 3. Verification of ASH-S Implementation
>
> - Due to the poor performance of ASH-S, we verified the results using the official code. Although the paper does not report ViT results, related scripts are available in their codebase. We tested the official [ViT script](https://github.com/andrijazz/ash/blob/main/config/vit_config.yml), and the results are shown below, confirming that ASH-S continues to perform poorly on ViT.
>
>     | ViT-B/16 | Score| iNaturalist FPR↓ | iNaturalist AUC↑ | SUN FPR↓ | SUN AUC↑ | Places FPR↓ | Places AUC↑ | Texture FPR↓ | Texture AUC↑ | Average FPR↓ | Average AUC↑ |
>     |:-|:-|-:|-:|-:|-:|-:|-:|-:|-:|-:|-:|
>     |Vanilla*| Energy |64.08 |79.24|72.77|70.25|74.30|68.44|58.46|79.30|67.40|74.31|
>     |ASH-S*| Energy |99.98 |7.28 |99.64|17.82|99.59|19.72|98.09|27.31|99.32|18.03|
>
>     > \* denotes our rerun
>
> **References**:
>
> - [1] Dice: leveraging sparsification for out-of-distribution detection.
> - [2] Extremely simple activation shaping for out-of-distribution detection.
> - [3] ReAct: out-of-distribution detection with rectified activations.
> - [4] Non-Parametric Outlier Synthesis.
> - [5] CLIP-driven Outliers Synthesis for few-shot OOD detection.

---

> > ### Comment · Reviewer_SyYP · 2024-08-12
> > **Thanks for the responses from the authors**
> >
> > Thanks for the responses from the authors.
> >
> > Q1. I think it is a crucial weakness that the method heavily depends on the loss (Eq. 11).
> >        Most previous research conducts experiments on CNNs.
> >        Although the authors provide comparisons with previous methods using the Swin backbone, the results seem to be a little bit wired, especially for ASH-S. I still recommend the authors could compare their method and previous works on CNNs for fair comparisons.

---

> ### Author Response · Authors · 2024-08-12
> **Results on CNNs**
>
> We sincerely appreciate the reviewer’s insightful comments. Here are our detailed responses:
>
> # 1. Results on ResNet50
>
> 1. **Setup:** We conducted experiments using only the ID loss, applying low-rank approximation on the in-feature and out-feature dimensions of the convolutional layers, combined with ASH for search. The results are as follows:
>
>     | ResNet50 |iNaturalist FPR↓ |iNaturalist AUC↑ |SUN FPR↓ |SUN AUC↑ |Places FPR↓ |Places AUC↑ |Texture FPR↓ |Texture AUC↑ |Average FPR↓ |Average AUC↑ |
>     |:--|---:|---:|---:|---:|-:|-:|--:|--:|--:|--:|
>     | Softmax \* |54.99|87.74 |70.83 |80.86 |73.99 |79.76 | 68.00 | 79.61 | 66.95 | 81.99 |
>     | Energy \* |55.72|89.95 |59.26 |85.89 |64.92 |82.86 | 53.72 | 85.99 | 58.41 | 86.17 |
>     | ReAct \* |20.38|96.22 |24.20 |94.20 |33.85 |91.58 | 47.30 | 89.80 | 31.43 | 92.95 |
>     | DICE \*|25.63|94.49 |35.15 |90.83 |46.49 |87.48 | 31.72 | 90.30 | 34.75 | 90.77 |
>     | ASH-P \* |44.57|92.51 |52.88 |88.35 |61.79 |61.79 | 42.06 | 89.70 | 50.32 | 89.04 |
>     | ASH-B \* |14.21|97.32 |22.08 |95.10 |33.45 |92.31 | 21.17 | 95.50 | 22.73 | 95.06 |
>     | ASH-S \* |11.49|97.87 |27.98 |94.02 |39.78 |90.98 | 11.93 |97.60| 22.80 | 95.12 |
>     | SeTAR|10.08|98.11|27.68|94.15|39.22|91.24| 12.54 | 97.51 | **22.38** | **95.25**|
>
>     > \* cite from "Extremely simple activation shaping for out-of-distribution detection."
>
> 2. **Not Highly Dependent on OOD Loss (Eq. 11):** Even under this simple setup, SeTAR outperforms ASH's best performance. This demonstrates that our method can effectively enhance the model's ability to distinguish between ID and OOD samples, even when relying solely on the ID loss. It’s important to note that this result was obtained under a basic setup, and due to time constraints, we did not further tune or explore other detailed configurations specific to CNNs, such as:
>
>    - **Low-Rank Settings for Convolutional Layers:** The optimal low-rank structure and dimensions for convolutional layers have not been thoroughly researched. For example, ELRT [1] proposes low-rank approximation directly in the high-order tensor format, while other methods [2][3][4] conduct and maintain the 4-D convolutional layer in the format of a low-rank 2-D matrix.
>
>    - **Pseudo-OOD Feature Extraction in CNNs:** As mentioned in our previous response, there are methods to construct OOD features within CNNs as well, which could further improve our model's performance.

---

> > ### Comment · Reviewer_SyYP · 2024-08-13
> >
> > Thanks for your responses to address my concerns.
> >
> > I strongly recommend including the comparisons with CNNs in the main paper. Although the improvements seem not obvious with CNNs, I still think the paper deserves to be accepted for its applications on CLIP models.

---

> > > ### Author Response · Authors · 2024-08-13
> > > **Grateful for Your Continued Feedback and Consideration**
> > >
> > > We sincerely appreciate the reviewer’s continued engagement and thoughtful feedback. We will include the CNN comparison results in the main paper as suggested.
> > >
> > > We hope that the additional comparisons enhance the overall evaluation of our work and would be grateful if the reviewer could kindly reconsider the rating in light of these updates.

---

> > > ### Author Response · Authors · 2024-08-14
> > > **Appreciation for Feedback and Request for Score Review**
> > >
> > > Dear Reviewer SyYP,
> > >
> > > Thank you for your recognition of our paper. We appreciate your comment that “the paper deserves to be accepted for its applications on CLIP models.” Your feedback has been incredibly valuable to us.
> > >
> > > As the rebuttal period is nearing its end, we have provided a quick summary of our responses and updates. We kindly ask you to consider these in your final scoring.
> > >
> > > Thank you once again for your valuable review.
> > >
> > > Best regards,
> > >
> > > The Authors

---

> ### Author Response · Authors · 2024-08-12
> **Results on CNNs (Continued)**
>
> # 2. Effectiveness of Our Method Across CLIP, Swin, and CNN Architectures
>
> 1. **Broad Effectiveness Across Architectures:** Our method has consistently proven effective across a range of architectures. As detailed in Table 1 and Table 7 of our paper, **SeTAR outperforms vanilla methods on both CLIP and Swin models**. Furthermore, even with a basic setup on ResNet, **SeTAR surpasses current state-of-the-art sparsification-based methods**. The comparison with Swin in our previous response underscores the limitations of previous sparsification-based approaches, which struggle with models like Swin. In contrast, **SeTAR achieves state-of-the-art performance across all major vision architectures**, demonstrating its versatility and generalizability.
>
> 2. **Significance for ViT-based Models:** ViT-based models, like CLIP, are receiving increasing attention in research due to their scalability and strong performance [5][6][7][8][9]. Our method’s superior results on ViT models highlight its potential for advancing OOD detection in these architectures, making it particularly relevant as ViT models become more widely adopted.
>
> 3. **Limitations of Sparsification-Based CNN Methods:** Sparsification-based CNN methods like ReAct and ASH cannot be applied as a post-hoc method to CLIP-based zero-shot OOD detection model. Both methods rely on the assumption that ID and OOD images produce distinct activations in models trained specifically on ID data, such as ResNet50. However, in large-scale pretrained models like CLIP, the activations for ID and OOD images are not significantly different. Consequently, methods like ReAct and ASH, which are limited to models trained on downstream ID-domain tasks, constrain their effectiveness in enhancing CLIP’s zero-shot OOD detection capabilities. In contrast, our method can be applied as a post-hoc method to enhance CLIP's zero-shot OOD detection capabilities.
>
> 4. **Contribution of Specialized Methods:** Despite their limitations, specialized methods play a crucial role in their respective domains. For instance, sparsification-based CNN methods significantly enhance OOD detection in CNN models, even though they may not perform well on CLIP models. Similarly, methods like GL-MCM [10] and LoCoOp [11], which utilize CLIP’s local features, substantially improve MCM scores and the performance of fine-tuned models. Although these methods are specialized, they contribute meaningfully to the ongoing development and advancement of the field.
>
> ---
>
> **References:**
>
> - [1] ELRT: Towards Efficient Low-Rank Training for Compact Neural Networks
> - [2] Learning Low-rank Deep-Neural Networks via Singular Vector Orthogonality Regularization and Singular Value Sparsification
> - [3] Training CNNs with Low-Rank Filters for Efficient Image Classification
> - [4] Convolutional Neural Networks with Low-Rank Regularization
> - [5] Multimodal Learning with Transformers: A Survey
> - [6] Self-Supervised Multimodal Learning: A Survey
> - [7] The Llama 3 Herd of Models
> - [8] Scaling Vision Transformers to 22 Billion Parameters
> - [9] How Far Are We to GPT-4V? Closing the Gap to Commercial Multimodal Models with Open-Source Suites
> - [10] Zero-Shot In-Distribution Detection in Multi-Object Settings Using Vision-Language Foundation Models
> - [11] LoCoOp: Few-Shot Out-of-Distribution Detection via Prompt Learning

---

### Official Review · Reviewer_aAbP · 2024-07-07

**Soundness:** 4
**Presentation:** 4
**Contribution:** 4
**Rating:** 6
**Confidence:** 4

**Summary:**

The paper presents SETAR, a novel method designed to enhance out-of-distribution (OOD) detection without requiring additional training. The proposed method leverages rank reduction techniques applied to the model weights, specifically targeting the minor singular components, while retaining the principal components that significantly contribute to the model’s performance. SETAR is evaluated across various model backbones, including CLIP-base, CLIP-large, and Swin-base, and demonstrates notable improvements in OOD detection tasks. The paper also provides comprehensive experiments and ablation studies to validate the effectiveness and efficiency of SETAR.

**Strengths:**

1.	Novelty and Innovation: The introduction of a training-free method for improving OOD detection is a significant contribution. By focusing on rank reduction of model weights, the method offers a fresh perspective compared to traditional training-intensive approaches.
	2.	Comprehensive Evaluation: The paper provides extensive experimental results across multiple datasets and model backbones, showcasing the robustness and generalizability of SETAR.
	3.	Effective Performance: The method achieves substantial improvements in OOD detection metrics, such as FPR95 and AUROC, demonstrating its practical utility. The significant performance boost on Swin-base compared to CLIP-base and CLIP-large is particularly notable, likely due to the inherent design differences and stronger zero-shot performance of CLIP models.
	4.	Detailed Analysis: The inclusion of ablation studies and sensitivity analyses helps in understanding the impact of various components and hyperparameters, offering valuable insights into the method’s functioning.

**Weaknesses:**

I hope the authors could address my following questions.

**Questions:**

1. Table 2: Could you provide more insights on why the performance boost on Swin-Base is significantly more pronounced compared to CLIP-Base and CLIP-Large? Is this disparity related to CLIP’s stronger zero-shot performance?

2. Figure 2: For ImageNet-1K, it appears that decomposing the vision encoder alone is sufficient for OOD detection, with the combination of both vision and text encoders yielding only minor improvements. Could you explain this in more detail?

3. The proposed method is training-free, but the greedy search algorithm used for determining the rank reduction ratio list and performing rank reduction does require computation time. Can you provide details on the time required for the greedy search and rank reduction for SeTAR, as well as the overall computation time of SeTAR+FT, compared to previous fine-tuning methods?

4. The proposed method seems to be an application of LASER, the only contribution is the adaptation of LASER to CLIP model, and benefits OOD detection. Can the authors justify this? The paper is technically solid with comprehensive experiments, but I think the paper is not novel enough given the similarities compared to LASER.

**Limitations:**

Limitation discussed in Appendix

---

> ### Author Rebuttal · Authors · 2024-08-07
>
> # 1. Performance Disparity on Swin-Base
>
> - The performance boost on Swin-Base is more pronounced because Swin is trained directly on ImageNet, lacking a text-encoder and thus requiring training solely on IN1K. This can lead to overfitting on ID data and poor recognition of OOD images, providing more room for improvement. Our method helps alleviate this issue, improving Swin's generalization to OOD samples.
>
> - In contrast, CLIP models are pretrained on large image-text datasets, which provides robust representation capabilities for both ID and OOD images. Consequently, CLIP’s baseline OOD performance is already strong, leaving less room for further improvement compared to Swin-Base. Therefore, the pronounced performance boost seen with Swin-Base is due to its initial lower performance and higher potential for enhancement through our method.
>
> # 2. Modality Difference for Performance
>
> 1. **Significance of Vision Modality:** As shown in Figure 2, the experiments demonstrate that the vision encoder is more critical for OOD detection tasks compared to the text encoder. This is intuitive since, in image-based OOD detection, the vision encoder is essential for extracting features and identifying OOD patches. The text encoder, on the other hand, shows limited improvement in performance for this specific task, indicating that the vision modality holds greater importance.
>
> 2. **Combined Modality for Optimal Performance:** While decomposing the vision encoder alone provides substantial improvements, incorporating the text encoder, albeit with minor gains, ensures we maximize the model’s capabilities. By leveraging both vision and text modalities, we achieve better overall performance, which is why we opted for the Vision+Text modality approach in our method.
>
> # 3. Computation Time Concerns for SeTAR and SeTAR+FT
>
> We appreciate the reviewer's concern regarding the computation time required for our proposed SeTAR method. Here is the comparison for CLIP-base on ImageNet1K:
>
> 1. **SeTAR and Fine-Tuning Times:**
>    - SeTAR requires approximately 14 minutes to complete the Vision+Text greedy search and rank reduction for 1K images. If we only apply the search to the Vision modality, the total time is about 7 minutes, which also achieves competitive performance as noted in the previous point.
>    - SeTAR+FT takes a total of 14 minutes and 11 seconds, consisting of two stages. The first stage is SeTAR for low-rank searching, which takes about 14 minutes. The second stage is LoRA tuning, which takes around 11 seconds for 5 epochs on the same 1,000 images, primarily due to the small model size and limited development set.
>    - For comparison, LoCoOp fine-tuning takes about **16 minutes** for 50 epochs.
>
> 2. **Detailed Time Analysis for SeTAR:**
>    - **Greedy Search Loss Calculation:** This accounts for 47.67% of the total time.
>    - **Dataloader:** Takes up 11.24% of the time. This delay occurs because the CPU cannot match the GPU speed for smaller models and few samples; for larger models, this delay can be negligible.
>    - **SVD Pruning and Reloading:** To maintain code clarity and compatibility, each step involves reloading and applying SVD pruning, which takes 20% of the total time. We plan to optimize this by loading the model once at the beginning and pre-computing SVD in parallel to avoid redundant calculations. This optimization could reduce the time taken by these steps to about 1/24 of its current value.
>    - With these optimizations, we estimate the overall time for SeTAR could be reduced to about half of the current duration, approximately **7 minutes** for Vision+Text and **3.5 minutes** for Vision-Only searching.
>
> These points highlight the efficiency of SeTAR and SeTAR+FT in both low-rank searching and fine-tuning compared to existing methods.
>
> # 4. Novelty Concerns Compared to LASER
>
> We appreciate the reviewer's feedback and the opportunity to clarify the unique contributions of SeTAR compared to LASER:
>
> 1. **Distinct Approach of SeTAR:**
>    - **Beyond Simple Application:** SeTAR is not just an adaptation of LASER to the CLIP model. LASER primarily focuses on pruning individual layers to enhance factual answering capabilities and does not extensively explore different greedy pruning strategies. Additionally, LASER relies on a validation set for selection, which is not suitable for OOD detection.
>    - **Greedy Pruning Algorithm and OOD Detection:** SeTAR focuses on designing a greedy pruning algorithm tailored for OOD detection. To address the challenge of unavailable OOD validation sets, we extract OOD information from ID images to guide the algorithm. We also conducted a sensitivity analysis of different parameters. Furthermore, SeTAR includes a comprehensive analysis comparing various modalities, search algorithms, pruning strategies, and backbones, enhancing our understanding of low-rank pruning beyond what LASER provides.
>
> 2. **Innovations in SeTAR+FT:**
>    - **Dynamic Rank Adjustment:** In addition to the training-free search algorithm, we explored the potential of using SeTAR for fine-tuning. Traditional LoRA distributes the rank evenly across all layers, leading to inefficiencies and performance losses.
>    - **Effective and Efficient Fine-Tuning:** By combining SeTAR with LoRA, SeTAR first identifies the impact of different layers on performance and dynamically adjusts the rank accordingly. This approach initializes different LoRA weights more effectively, tailored to the specific ID dataset, resulting in a more effective and efficient fine-tuning process.
>
> These points illustrate that SeTAR offers significant advancements over LASER, both in methodology and application, particularly for OOD detection.

---

> > ### Comment · Reviewer_aAbP · 2024-08-10
> >
> > Thank you to the authors for providing comprehensive clarifications. Most of my concerns have been addressed. I would like to note that the contribution of greedy rank search across multiple layers has already been proposed in the LASER paper (Sec. 5.1, “Composing reductions across layers”). Given the technical soundness, extensive experiments, and the innovative application of rank reduction in the domain of OOD detection using a Vision-Language model, I will maintain my original score.

---

> ### Author Response · Authors · 2024-08-12
> **Clarification and Innovations in SeTAR for OOD Detection**
>
> We sincerely appreciate the reviewer’s thoughtful feedback and the opportunity to clarify and expand on certain aspects of our work.
>
> # 1. Clarification on LASER
>
> We appreciate the reviewer’s reminder regarding Section 5.1 of the LASER paper. We also took note of this section during our review. What we intended to highlight in our original rebuttal is that, while LASER employs a single greedy search strategy, our work delves deeper into the nuances of conducting greedy search effectively for OOD detection (as detailed in Section 4.4). Specifically, LASER’s approach focuses on composing reductions across layers without exploring the broader landscape of greedy search strategies. In contrast, we systematically analyze and compare different greedy search techniques, evaluating their effectiveness across various layers and backbones. This detailed exploration allows our method to be more finely tuned for the specific challenges of OOD detection, thereby providing a more robust and versatile solution.
>
> # 2. Innovation in Post-Hoc Sparsification Methods
>
> Post-hoc sparsification methods are widely utilized to enhance CNN-based OOD detection, with well-known examples including ReAct[1], ASH[2], and Dice[3]. These methods typically operate in the weight or activation space, aiming to improve OOD detection by modifying the model’s internal structures. However, a significant limitation of methods like ASH and ReAct is that they are designed for models that have been trained on in-domain (ID) data, such as ResNet50, where distinct activations for ID and OOD samples are expected. These methods fail to generalize to zero-shot OOD detection models like CLIP, where the model has not been fine-tuned on any ID data, and the activations for ID and OOD samples share similar distributions.
>
> Our approach addresses these limitations by introducing two key innovations: first, we operate within the space derived from SVD decomposition, which allows us to capture the most informative components of the model while discarding noise. Second, our method is specifically designed to function as a post-hoc approach compatible with CLIP’s zero-shot OOD detection capabilities. This dual innovation not only enables our method to bypass the limitations of traditional sparsification techniques but also allows it to enhance OOD detection in models that are not specifically trained on ID data. We will ensure that these distinctions and innovations are clearly articulated in the revised version of our paper.
>
> ---
>
> **References**:
>
> - [1] ReAct: out-of-distribution detection with rectified activations.
> - [2] Extremely simple activation shaping for out-of-distribution detection.
> - [3] Dice: leveraging sparsification for out-of-distribution detection.

---

> ### Comment · Reviewer_aAbP · 2024-08-13
>
> Thank you for your additional clarifications. Based on your responses to all the reviewers, I think the paper’s contributions are sufficient for acceptance. I recommend that the authors expand the discussion of related works on OOD Detection in Computer Vision [1,2,3] in the revised version to further strengthen the paper. I will raise my score to 6.
>
> [1] How to Exploit Hyperspherical Embeddings for Out-of-Distribution Detection?
>
> [2] Learning with Mixture of Prototypes for Out-of-Distribution Detection
>
> [3] Energy-based Hopfield Boosting for Out-of-Distribution Detection

---

> > ### Author Response · Authors · 2024-08-13
> > **Gratitude for Your Valuable Feedback and Support**
> >
> > We sincerely appreciate the reviewer’s thoughtful comments and the decision to raise the score. We will ensure that the discussion on related works in OOD detection, particularly the references provided, is expanded and integrated into the revised version of our paper. This will further strengthen the context and positioning of our contributions. Thank you again for your valuable feedback and support throughout the review process.

---

### Author Response · Authors · 2024-08-14
**General Responses**

**Dear AC and Reviewers,**

We thank all reviewers for their thorough comments and constructive feedback. We greatly appreciate that the reviewers found our paper to be well-written (SyYP, hoZJ), novel (aAbP, 6fvA), effective (SyYP, aAbP), with solid and comprehensive experiments and analysis (aAbP, 6fvA), and sufficient for acceptance (SyYP, aAbP). We would like to summary some key points in the rebuttal:

**1. Effectiveness of Our Method:**
SeTAR consistently demonstrates superior performance across various OOD detection tasks, surpassing existing zero-shot methods and sparsification techniques. This includes its effectiveness when applied to different score functions and backbones, such as CLIP, Swin, and ResNet. Notably, SeTAR+FT outperforms the current state-of-the-art in efficient fine-tuning baselines, offering significant improvements in both AUC and FPR metrics. In scenarios involving label expasion, SeTAR not only matches but exceeds the performance of the baseline NegLabel, establishing a new state-of-the-art. Furthermore, our method exhibits robust performance in both Far-OOD and Near-OOD scenarios, making it a versatile and powerful tool for diverse OOD detection challenges.

**2. Compatibility:**
A key strength of SeTAR is its high compatibility with various score functions (MCM, GL-MCM, MSP, Energy), multiple model backbones (CLIP, Swin, ResNet), and advanced OOD methods like CLIPN and NegLabel. This flexibility distinguishes SeTAR from other CLIP-based methods such as CLIPN, NegLabel, and LoCoOp, as well as from CNN-based sparsification methods. The ability to integrate seamlessly with different approaches underscores SeTAR's potential to enhance and complement existing methods, rather than being in conflict with them. This compatibility allows our method to be effectively applied in a wide range of OOD detection settings, further solidifying its practical value.

**3. Efficiency:**
SeTAR is designed to be both lightweight and efficient, addressing the need for a resource-effective solution in OOD detection. Specifically tailored for zero-shot scenarios, SeTAR requires no additional training, making it highly practical for real-world applications where computational resources may be limited. Moreover, SeTAR+FT leverages our approach alongside existing parameter-efficient fine-tuning methods, enabling highly efficient training while still achieving state-of-the-art performance. This efficiency is particularly evident when compared to methods that rely heavily on extensive training or large amounts of external data, such as CLIPN or NegLabel, which require significantly more resources.

We are committed to refining our paper to ensure it is as clear and accurate as possible. We will make sure to address all the comments and suggestions provided by the reviewers and will expand the discussion of related works on OOD detection in computer vision to strengthen the context and positioning of our contributions. We sincerely appreciate the reviewers' insights and support throughout this process.

Sincerely,

All authors

---

### Decision · Program_Chairs · 2024-09-25

**Decision:**

Accept (poster)

**Comment:**

All three reviewers agree that this work is of interest and value to the NeurIPS community and that it should be accepted. The AC agrees.

Authors are requested to revise their manuscript based on the reviewer feedback, incorporating the various discussions and additional results from the author-reviewer exchanges.